# Dysfunction of ventrolateral striatal dopamine receptor type 2-expressing medium spiny neurons impairs instrumental motivation

Iku Tsutsui-Kimura[1,2,*], Hiroyuki Takiue[1,3,*], Keitaro Yoshida[1], Ming Xu[1], Ryutaro Yano[1,3], Hiroyuki Ohta[4], Hiroshi Nishida[1], Youcef Bouchekioua[1], Hideyuki Okano[3], Motokazu Uchigashima[5], Masahiko Watanabe[5], Norio Takata[1], Michael R. Drew[6], Hiromi Sano[7], Masaru Mimura[1] & Kenji F. Tanaka[1]

Impaired motivation is present in a variety of neurological disorders, suggesting that decreased motivation is caused by broad dysfunction of the nervous system across a variety of circuits. Based on evidence that impaired motivation is a major symptom in the early stages of Huntington's disease, when dopamine receptor type 2-expressing striatal medium spiny neurons (D2-MSNs) are particularly affected, we hypothesize that degeneration of these neurons would be a key node regulating motivational status. Using a progressive, time-controllable, diphtheria toxin-mediated cell ablation/dysfunction technique, we find that loss-of-function of D2-MSNs within ventrolateral striatum (VLS) is sufficient to reduce goal-directed behaviours without impairing reward preference or spontaneous behaviour. Moreover, optogenetic inhibition and ablation of VLS D2-MSNs causes, respectively, transient and chronic reductions of goal-directed behaviours. Our data demonstrate that the circuitry containing VLS D2-MSNs control motivated behaviours and that VLS D2-MSN loss-of-function is a possible cause of motivation deficits in neurodegenerative diseases.

[1] Department of Neuropsychiatry, Keio University School of Medicine, Tokyo 160-8582, Japan. [2] Research Fellow of Japan Society for the Promotion of Science (RPD), Tokyo 102-0083, Japan. [3] Department of Physiology, Keio University School of Medicine, Tokyo 160-8582, Japan. [4] Department of Physiology, National Defense Medical College, Saitama 359-8513, Japan. [5] Department of Anatomy and Embryology, University of Hokkaido, Hokkaido 060-8638, Japan. [6] Center for Learning and Memory, Department of Neuroscience, The University of Texas at Austin, Austin, Texas 78712, USA. [7] Division of System Neurophysiology, National Institute for Physiological Sciences, Okazaki 444-8585, Japan. * These authors contributed equally to this work. Correspondence and requests for materials should be addressed to K.F.T. (e-mail: kftanaka@keio.jp).

Impaired goal-directed motivation is a common feature of many pathologies involving neural degeneration, lesion, or dysfunction, including Alzheimer's disease[1,2], Parkinson's disease[3], Huntington's disease[4,5], progressive supranuclear palsy[6] and stroke[7,8]. Reduced motivation, one of the core symptoms of apathy[9,10], is associated with impairments in activities of daily living and poor response to rehabilitation. Therefore, understanding the mechanisms of decreased motivation in these disorders is of utmost importance.

Because impaired motivation occurs in a variety of disorders, it has been suggested that the deficits are produced by broad dysfunction, most likely within the cortico-striatal system, whose role in motivation has been established in animal studies[11]. However, the precise mechanisms through which neurodegeneration within the cortico-striatal system impairs motivation are not understood. To model neurodegeneration-associated decreased motivation and understand the pathogenesis in detail, we focused on two previous findings: (1) impaired motivation is the most common behavioural disturbance resulting from striatal lesion[12], (2) impaired motivation is a major neuropsychiatric symptom in the early stages of Huntington's disease, in which dopamine receptor type 2-expressing striatal medium spiny neurons (D2-MSNs) are particularly degenerated[13,14]. These findings encouraged us to address whether bilateral D2-MSN-specific ablation or dysfunction is sufficient to impair goal-directed motivation.

To test this hypothesis we established a progressive neurodegeneration mouse model based on targeted expression of diphtheria toxin in D2-MSNs. Using a variety of behavioural tasks, we establish that ablation or dysfunction of D2-MSNs specifically within the VLS is sufficient to reduce goal-directed motivation. We further show that goal-directed motivation can be impaired transiently or chronically via optogenetic inhibition or optogenetic ablation of VLS D2-MSNs. Our data implicate degeneration of VLS D2-MSNs, as a potential cause of motivation deficits in neurodegenerative diseases such as Huntingon's.

## Results

**Stepwise regional expansion of D2-MSN ablation.** To target D2-MSNs, we exploited the tTA-tetO system[15], in which a tTA was expressed under the control of the *Drd2* promoter (*Drd2*-tTA), and a DTA was induced in a DOX-dependent manner (tetO-DTA)[16] (Fig. 1a; Supplementary Fig. 1A–C). Bigenic animals (*Drd2*-tTA::tetO-DTA, hereafter referred to as D2-DTA) were fed with doxycycline (DOX)-containing chow until the start of the experiment. On switching to normal chow (DOX-off day 0), tTA-mediated DTA induction began. The time courses of histopathological events after DOX removal were as follows (Fig. 1b–f; Supplementary Fig. 1D,E): a few, scattered *DTA* messenger RNA (mRNA) signals were first detected in the bilateral ventrolateral striatum (VLS) at DOX-off day 3. The number of *DTA* mRNA-positive cells was increased at DOX-off day 7, but with no apparent loss of *Drd2* mRNA signal and with no cell death in the corresponding region. Regional loss of *Drd2* mRNA signal was first detected only in the VLS at DOX-off day 10, and a few single-strand DNA-positive dead cells were detected. *Drd2* mRNA-negative areas had expanded concentrically by DOX-off day 14 (VLS, ventromedial striatum (VMS), and ventral part of the dorsomedial striatum) (Fig. 1c,d, and Supplementary Fig. 2) when numerous dead cells were detected. Activation of glia coincided with cell death (Fig. 1e,f). The *Drd2* mRNA-negative area covered all of the striatum except for the farthest dorsolateral part at DOX-off day 28 (Fig. 1c,d). In summary, DTA-expressing (*DTA* mRNA-positive) cells were viable for several days, and then cell death occurred (*Drd2* mRNA signal disappearance coincided).

The numbers of *Drd2* mRNA-positive striatal cholinergic interneurons and dopaminergic neurons did not change after DOX removal (Fig. 1g,h,j). Indeed, after DOX removal, *DTA* mRNA was not detectable in cholinergic interneurons (Fig. 1i) and in dopaminergic neurons. In addition, *Drd2* mRNA-positive cells outside the basal ganglia, such as the pyramidal neurons in the cortex (Supplementary Fig. 3), did not express *DTA* mRNA after DOX removal, supporting the specificity of DTA expression in D2-MSNs (Supplementary Table 1). The number of dopamine receptor type 1-expressing medium spiny neurons (D1-MSNs) in the VLS did not change after DOX removal (Fig. 1k).

*DTA* mRNA expression initiated in the VLS was not limited to the rostral part of the striatum, which included the lateral part of the nucleus accumbens, but spread from the rostral to the caudal part of the striatum (Fig. 2a). The rostro-caudal cylindrical DTA-expression expanded progressively in a concentric manner (Supplementary Fig. 1D) followed by cylindrical *Drd2* mRNA-negative area expansion (Supplementary Figs 1E and 2). Importantly, these unique pathological changes and time courses were reliably reproduced within sets of transgenic animals (overall $n > 50$).

By taking advantage of stepwise expanding DTA-expression from ventral to dorsal subregions, we confirmed the prevailing findings that dorsal D2-MSNs control locomotor activity. Locomotor activity in an open field box was recorded for 30 min in D2-DTA bigenic mice with DOX off for 10, 14 and 28 days. Mice with DOX off for 10 days showed comparable locomotor activity to DOX-on D2-DTA bigenic littermate controls ($t_{35} = 0.922$, $P = 0.363$), whereas mice exhibited hyperactivity after DOX was off for 14 days ($t_{27} = 5.054$, $P < 0.001$) and 28 days ($t_{12} = 5.714$, $P < 0.001$) compared with controls (Fig. 2b), consistent with findings that that loss-of-function of dorsal (dorsomedial) D2-MSNs results in hyperactivity[17].

**DTA-exposed but viable D2-MSNs are hypofunctioning.** Our histological analysis revealed that cell death and apparent loss of *Drd2* mRNA occurred after DOX-off day 10. However, before cell death, *DTA* mRNA was expressed at earlier times (DOX off for 3–7 days) (Fig. 1b,f). To address whether viable DTA-expressing cells behaved normally or not, we examined the function of VLS D2-MSNs, which are thought to relay signals from the insular cortex (IC) to the ventral pallidum. Indeed, projection neurons from the IC terminate within the VLS (Allen Institute for Brain Science Mouse Connectivity Atlas, http://connectivity.brain-map.org/, experiments 174361746, Image 106, 95 and 86) and ventral striatum MSNs terminate within the ventral pallidum (VP)[18]. This circuit shares the similarity to prefrontal-medial accumbal-ventral pallidal connectivity[19].

When we electrically stimulated the IC and measured responses within the VP (Fig. 3a), three distinct responses were detected as early excitation, inhibition and late excitation (Fig. 3b). The early excitation was mediated via IC-subthalamic nucleus (STN) projections, which did not include the striatal projections. The inhibition following the early excitation resulted from the activation of a feedback inhibitory circuit that was mediated by VLS. The late excitation involved the IC-VLS-VP-STN pathway that operated via a disinhibitory process. Therefore, if MSN function were compromised, the degrees of inhibition and late excitation would be diminished.

We measured VP responses from bigenic DOX-on controls ($n = 5$, intact D2-MSNs), D2-DTA bigenic DOX-off day 7

($n = 3$, viable, DTA-exposed D2-MSNs) and DOX-off day 20 ($n = 4$, ablated D2-MSNs) animals. The rates of triphasic patterns significantly decreased in both D2-DTA bigenic mice at DOX-off day 7 and 20 compared with those in controls. (DOX-on, 22.4%, 73 out of 326 versus DOX-off 7, 3.2%, 13 out of 404: $P < 0.001$, $\chi^2$ test; DOX-on versus DOX-off 20, 0.6%, 2 out of 314: $P < 0.001$, $\chi^2$ test). Rates containing inhibitory and/or late excitatory responses also decreased (DOX-on, 76.1%, 248 out of 326 versus DOX-off 7, 31.9%, 129 out of 404: $P < 0.001$, $\chi^2$ test; control versus DOX-off 20, 47.7%, 149 out of 314: $P < 0.001$, $\chi^2$ test, Fig. 3d). The latency and the duration of inhibitory responses respectively increased and decreased in both DOX-off day 7 and 20 (DOX-on versus DOX-off 7: $P < 0.001$; DOX-on versus DOX-off 20: $P < 0.001$; multiple

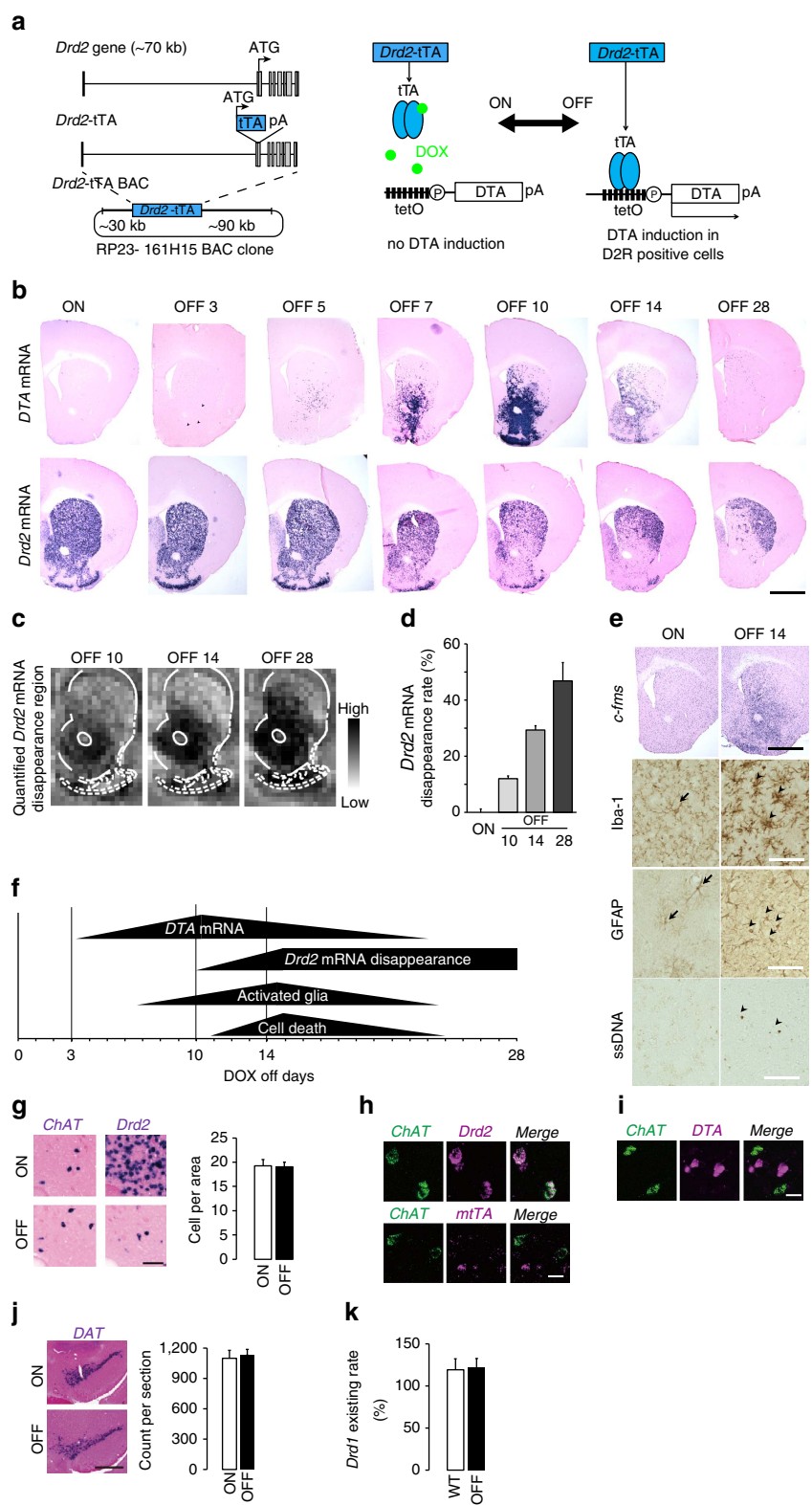

comparison with Bonferroni's method, Table 1), and the latency and the duration of late excitatory responses exhibited the same results as inhibitory responses (DOX-on versus DOX-off 7: $P < 0.001$; DOX-on versus DOX-off 20: $P = 0.052$; multiple comparison with Bonferroni's method, Table 1). Populations of peristimulus time histograms revealed blurred responses of inhibition and late excitation (Fig. 3e). Collectively, these data point to MSN hypofunction at both earlier and later time points off DOX treatment. In other words, living (viable) DTA-expressing D2-MSNs were hypofunctioning, indicating that the loss-of-function occurred before DTA-mediated cell death.

**VLS D2-MSN loss-of-function decreases motivation initially**. We asked whether the loss-of-function of bilateral VLS D2-MSNs induces decreased motivation. We combined a modified 3-choice serial reaction time task (3-CSRTT, Fig. 4a) with the day-by-day expanding loss-of-function that was initiated from the ventrolateral D2-MSNs. This instrumental task required mice to perform a food-reinforced goal-directed behaviour, in which various parameters including motivation, attention, impulsivity, compulsivity, motor function and appetite can be addressed simultaneously and independently[20,21].

We initiated the DOX-off regimen after training. Following DOX-off conditions for 3 days, D2-DTA bigenic mice displayed a decreased total number of trials in 60 min of testing (phase from days 3 to 10: $F_{2,15} = 9.102$, $P = 0.004$; phase × group interaction: $F_{2,20} = 4.832$, $P = 0.022$, Fig. 4b) compared with monogenic controls (total trial: $t_{10} = 2.422$, $P = 0.045$, Fig. 4b). D2-DTA bigenic mice at DOX off day 10 showed normal locomotor activity (Fig. 2b), supporting the idea that the decreased number of total trials achieved was caused by impairment of instrumental motivation[20]. There was a trend of increased %omission (phase: $F_{2,15} = 2.943$, $P = 0.056$, NS, phase × group interaction: $F_{2,20} = 1.954$, $P = 0.061$, NS, Fig. 4c), which was likely because of the reduction of sustained motivation, rather than reduction of sustained attention, since other parameters representing cognitive activities were intact (%accuracy, phase: $F_{2,15} = 2.692$, $P = 0.095$, NS, phase × group interaction: $F_{2,20} = 0.301$, $P = 0.731$, NS, Fig. 4d; correct response latency, phase: $F_{2,15} = 1.992$, $P = 0.089$, NS; phase × group interaction, $F_{2,20} = 0.699$, $P = 0.493$, NS, Fig. 4e) in this study. Given these alterations in behaviour, loss-of-function of ventrolateral D2-MSNs induces quantitative reductions in goal-directed behaviour, which can be interpreted as decreased instrumental motivation. Mice experiencing the DOX-off day 7 regimen displayed a comparable food preference/intake (Supplementary Fig. 4), strengthening the selective effect of loss-of-function of D2-MSN on food-incentive instrumental tasks.

After a 14-day DOX-off regimen, at which point DTA-mediated ablation extended to the VMS, D2-DTA bigenic mice displayed increased impulsivity (%premature response, phase: $F_{2,15} = 10.633$, $P = 0.001$; group: $F_{1,10} = 14.512$, $P = 0.005$; phase × group interaction: $F_{2,20} = 4.772$, $P = 0.026$, Supplementary Fig. 4D), as well as increased compulsivity (%perseverative response, phase: $F_{2,15} = 8.375$, $P = 0.002$; phase × group interaction: $F_{2,20} = 9.744$, $P = 0.02$, Supplementary Fig. 4E), while the monogenic controls did not (%premature response: $t_{10} = 4.603$, $P = 0.002$, Supplementary Fig. 4D; %perseverative responses: $t_{10} = 3.148$, $P = 0.015$, Supplementary Fig. 4E). These results support the widely accepted theory that D2-MSNs in the VMS play a role in behavioural inhibition[22,23]. At this time point, indices of motivation were normalized (Fig. 4b,c), which was perhaps because of a masking effect by hyperactivity (Fig. 2b) or increased impulsivity (Supplementary Fig. 4D). These results indicate that the continuous DOX-off regimen produced decreased motivation transiently, while behaviours related to decreased motivation disappeared as the DTA-expressing area expanded outside the VLS.

**DTA-mediated cell ablation causes chronic decreased motivation**. To better discern the cellular population responsible for decreased motivation, we used the progressive ratio (PR) task[24] (Fig. 5a), which permitted us to address effort-related motivation simply and directly. In addition, we restarted DOX after 7 days of DOX-off conditions in order to cease further expansion of DTA mRNA induction and to restrict the cell dysfunction/ablation within the VLS. D2-DTA bigenic mice started to display a behavioural reduction after DOX was off for 3 days and this reduction further deteriorated day-by-day according to decreased break points (phase × group interaction: $F_{2,40} = 5.782$, $P = 0.021$, Fig. 5b) and prolonged time spent to complete the PR task (phase × group interaction: $F_{2,40} = 7.344$, $P = 0.003$, Fig. 5c). These observations were not evident in controls (break point: $t_{20} = 13.211$, $P = 0.005$, time spent to complete the PR: $t_{20} = 15.899$, $P = 0.002$, Fig. 5b,c). After the DOX restart, the behavioural reduction remained (break point, *post hoc* analysis between groups at phase III: $t_{20} = 17.377$, $P = 0.004$, Fig. 5b; time spent to complete the PR, *post hoc* analysis between groups at phase III: $t_{20} = 21.093$, $P = 0.002$, Fig. 5c). Associative learning and appetite were unaffected (Fig. 5d,e) as seen in the 3-CSRTT (Fig. 4d,e), suggesting that cognitive and emotional dimensions were spared. These data suggest that the combination 7-day

**Figure 1 | Time-controllable diphtheria toxin-mediated D2-MSN-specific loss-of-function manipulation.** (**a**) *Drd2*-tTA BAC transgene (left). The tTA complementary DNA (cDNA) was inserted into the start codon of the *Drd2* gene. DOX-controllable DTA expression (right). *DTA* mRNA expression started when DOX-chow was replaced with normal chow (DOX-OFF). The DOX restart terminated *DTA* mRNA induction (DOX-ON). (**b**) *DTA* mRNA (upper) and *Drd2* mRNA expression (lower) in the striatum in each DOX-off period. Purple colour denotes the mRNA signal. *DTA* mRNA expression was observed in the VLS, with levels and areas increasing, then decreasing by day 28 off DOX. *Drd2* mRNA disappearance area increased after 10 days off DOX. Scale,1 mm. (**c**) Averaged *Drd2* mRNA disappearance areas ($n = 5$) in each DOX-off period. The darker grey areas indicate greater *Drd2* mRNA disappearance. (**d**) Quantification of *Drd2* mRNA disappearance rate in each DOX-off period. (**e**) DTA-mediated cell death and concurrent glial activation. Left, control; right, D2-DTA at 14 days off DOX. Arrows show healthy glia. Glial activation (arrowheads) coincided with cell death. *c-fms* and Iba1: microglia marker, GFAP: astrocyte marker. Single-strand DNA (ssDNA)-positive dead cells were found mostly in the VLS (arrowheads). Black scale, 1 mm. White scale,50 μm. (**f**) A schematic timeline of histopathology in the VLS. Note the time lag of *Drd2* mRNA loss. (**g**) Cholinergic interneurons in the striatum were spared in D2-DTA mice at 14 days DOX-off (left). Scale = 100 μm. No change of number of cholinergic interneurons after DOX-off (right, $n = 8$ for each group, $t_{14} = 0.135$, $P = 0.894$) Error bars indicate s.e.m. (**h**) *ChAT* mRNA-positive cholinergic interneurons expressed *Drd2* mRNA (upper panels); however, *Drd2*-tTA mice failed to express *tTA* mRNA in *Drd2* mRNA-positive cholinergic interneurons (lower panels). Scale, 20 μm. (**i**) *ChAT* mRNA-positive cells were not labelled with *DTA mRNA*. Scale,20 μm. (**j**) The number of *Dopamine transporter (DAT)* mRNA-positive dopamine neurons did not change after DTA induction. ($n = 5$ for each group, $t_8 = 1.318$, $P = 0.224$). Scale, 500 μm. Error bars indicate s.e.m. (**k**) Quantification of relative *Drd1* mRNA expression rate within VLS in D2-DTA (DOX-off days 20) and control ($n = 4$, each).

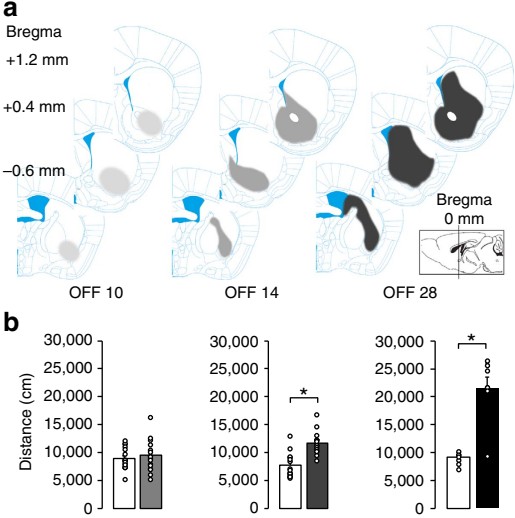

**Figure 2 | Anatomical expansion of *Drd2* mRNA disappearance area and behavioural readout.** (**a**) Stepwise expansion of *Drd2* mRNA disappearance from the rostral to the caudal striatum (grey area) is illustrated. (**b**) Locomotor activity in the open field test. Significant increases in total distance were observed in the 14- and 28-day groups, but not in the 10-day group. Bars represent the mean and lines represent the s.e.m. *$P < 0.05$, Student's *t*-test.

DOX-off and 14-day DOX-restart regimen succeeded to confine the lesion area within the VLS and to produce chronic neurodegeneration-associated decreased motivation, which was characterized by a reduction of self-generated goal-directed behaviour without cognitive or emotional impairments.

To address whether the degree of neuroanatomical loss-of-function correlated with the behavioural data, we quantified the loss of *Drd2* mRNA-positive cells at the endpoint of the 7-day DOX-off and 14-day DOX-restart regimen. We found that the degree of anatomical deficit (*Drd2* mRNA-negative area in the VLS, Supplementary Fig. 5A) was correlated with the degree of behavioural disruption ($r = 0.892$; $P = 0.017$, Fig. 5f), supporting the notion that D2-MSN ablation within the VLS causes decreased motivation. We also found that 17% of D2-MSNs in the VLS were positive for *DTA* mRNA at the 5-day DOX-off early time point (Supplementary Fig. 5B), suggesting that the manipulation of 17% of D2-MSNs in the VLS is sufficient to trigger decreased motivation.

**Opto-inhibition/ablation of VLS D2-MSNs impairs motivation.** To examine whether an acute loss-of-function of bilateral VLS D2-MSNs induces a reduction of goal-directed behaviours, we employed optogenetic inhibition of bilateral VLS D2-MSNs during the progressive ratio task. We generated bigenic animals in which D2-MSNs expressed archaerhodopsin[25] (*Drd2*-tTA::tetO-ArchT-EGFP, Fig. 6a; Supplementary Fig. 6A and B, hereafter referred to as D2-ArchT). *In vitro* electrophysiology demonstrated that the current injection-induced spike generation of ArchT expressing VLS MSNs was suppressed by yellow but not blue light illumination (Fig. 6b).

Immunohistochemistry detected a few GFP-positive cells at dopamine neurons (Supplementary Fig. 6C); however, the level of GFP was too low to observe direct fluorescence, suggesting that optogenetic inhibition should not work in dopamine neurons because of the low level of ArchT expression. *Drd2*-positive cholinergic interneurons were not labelled with

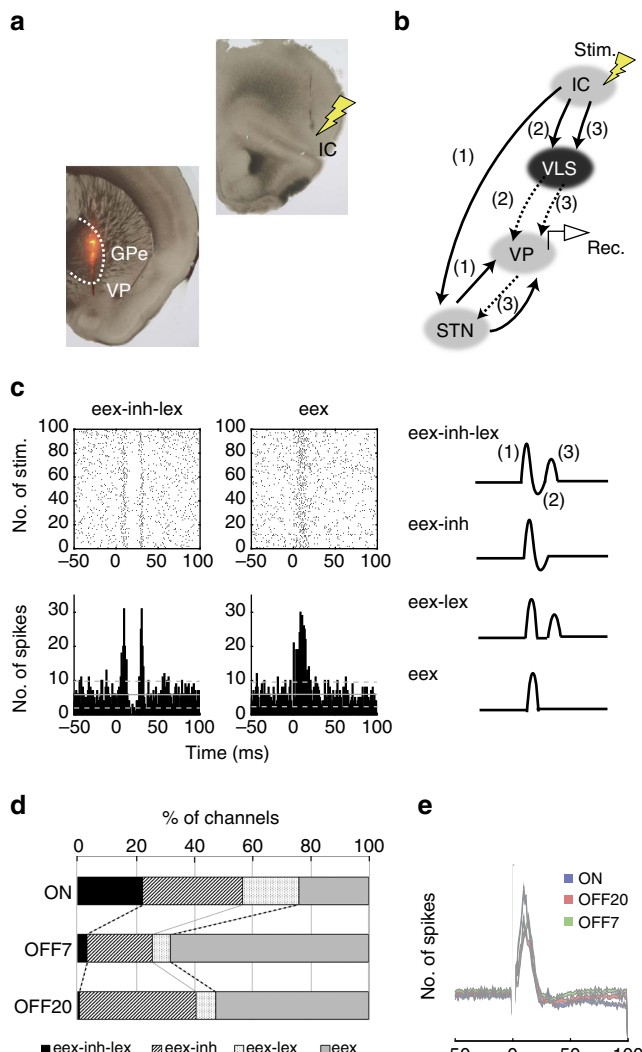

**Figure 3 | Electrophysiological characterization of DTA-exposed but viable D2-MSNs.** (**a**). Positions of stimulation (arrow at the insular cortex (IC) and recording (red) are shown. Note that the ventral pallidum (VP) is located beneath the globus pallidus externa (GPe) in this AP position. (**b**) Schematic representation of the pathways from the stimulation site to the recording site. Pathway 1 (IC-subthalamic nucleus (STN) -VP) drives the early excitation; pathway 2 drives inhibition (IC-VLS-VP); pathway 3 (IC-VLS-VP-STN-VP) drives the late excitation. Solid and dotted lines represent glutamatergic and GABAergic pathways, respectively. (**c**) Raster and PSTHs of the typical patterns of eex-inh-lex and eex of VP neurons. Cortical stimulation (200 µs duration single pulse, 200 µA strength) was delivered at time 0 for 100 stimulus trials. The mean firing frequency and statistical levels of $P < 0.05$ (one-tailed *t*-test) calculated from the 100 ms period preceding the onset of stimulation are indicated by solid lines (mean) and dotted lines (statistical levels), respectively. Four patterns (eex-inh-lex, eex-inh, eex-lex, eex) were classified. eex indicates early-excitation; inh, inhibition; lex indicates late-excitation. (**d**) Proportions of VP neurons (% of traces) classified according to the response patterns evoked by the IC stimulation in controls and D2-DTA DOX-off mice. % of VP neurons with inh and/or lex (patterns with eex-inh-lex, eex-inh and eex-lex in total) was lower in D2-DTA DOX-off mice. (**e**) Difference in the population PSTHs of VP neurons among D2-DTA DOX-on, DOX-off day 7, and DOX-off day 20 groups. The light-shaded colours represent ± s.e.m.

**Table 1 | Response parameters of VP neurons to insular cortical stimulation in D2-DTA mice at DOX-on, DOX-off day 7 and 20.**

| Number of traces | DOX-on 326 | DOX-off day 7 404 | DOX-off day 20 314 |
|---|---|---|---|
| *Early excitation* | $n = 326$ | $n = 404$ | $n = 314$ |
| Latency (ms) | 5.3 ± 0.1 | 8.0 ± 0.3* | 6.7 ± 0.2* |
| Duration (ms) | 11.9 ± 0.3 | 9.1 ± 0.4 | 9.1 ± 0.4 |
| *Inhibition* | $n = 185$ | $n = 104$ | $n = 124$ |
| Latency (ms) | 25 ± 0.4 | 29 ± 0.4* | 27 ± 0.4* |
| Duration (ms) | 8.0 ± 0.4 | 3.9 ± 0.2* | 5.2 ± 0.4* |
| *Late excitation* | $n = 136$ | $n = 38$ | $n = 23$ |
| Latency (ms) | 38 ± 0.7 | 48 ± 0.3* | 38 ± 0.5 |
| Duration (ms) | 5.7 ± 0.3 | 2.5 ± 0.1* | 2.3 ± 0.0* |

Note: Values are mean ± s.d. Values with an asterisk are significantly different from those of control ($P < 0.001$; multiple comparison with Bonferroni's method).

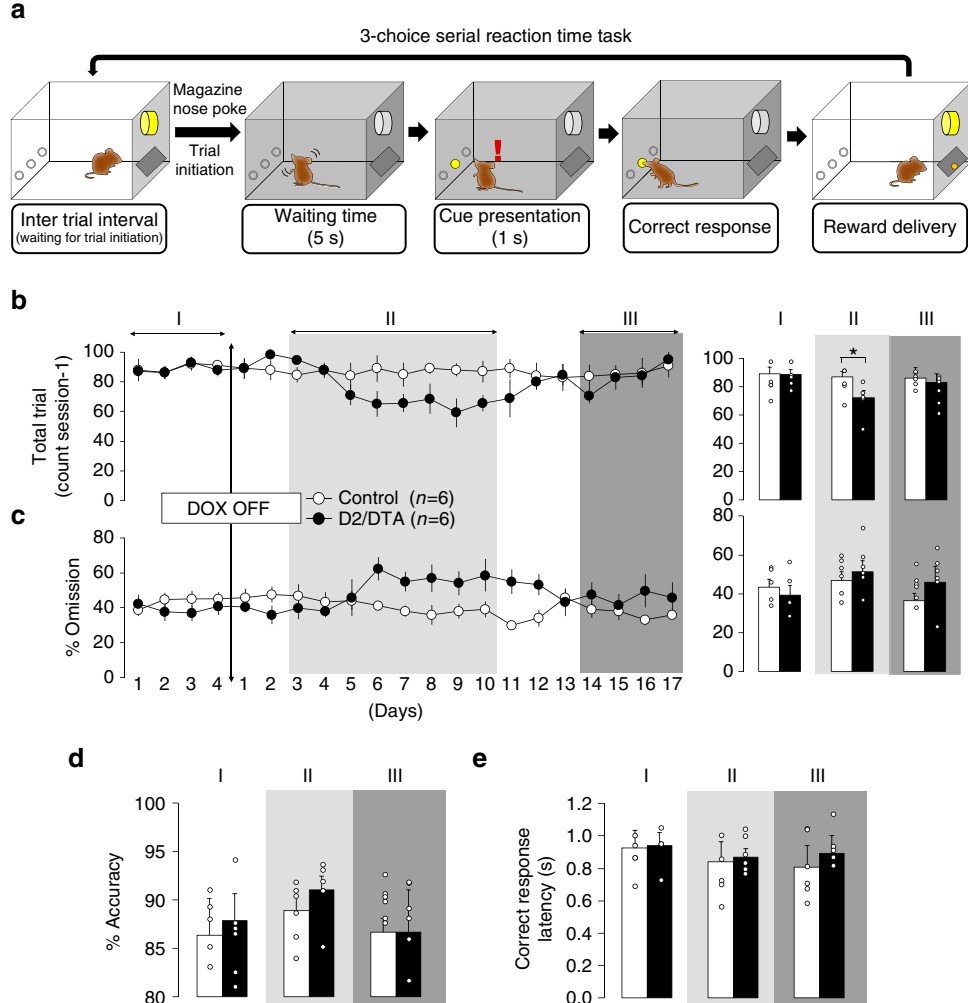

**Figure 4 | Expansion of VLS D2-MSN loss-of-function initially produces a reduction of goal-directed behaviour.** (**a**) Schematic illustrations of the 3-CSRT task sequence. (**b**–**e**) The temporal changes of behavioural parameters induced by DOX-off manipulations in the 3-CSRTT. The black circles represent data from the bigenic group ($n = 6$) and the white circles represent data from monogenic controls ($n = 6$). The bar graphs show the behavioural parameters averaged over −3 to 0 days before (phase I), 3–10 days after (phase II) and 14–17 days after (phase III) DOX termination in the 3-CSRTT. Decreased total number of trials (**b**) and a trend of increased % omissions (**c**) were observed only during phase II. DOX removal did not change (**d**) % accuracy or (**e**) correct response latency, indicating that divided attention and motor function were spared after DOX removal. Bars represent the mean and lines represent the s.e.m.

GFP (Supplementary Fig. 6D). The pyramidal neurons are known to express *Drd2* mRNA, and their axon terminals project to the striatum; however, neurons in the medial prefrontal cortex and

IC were not labelled with GFP (Supplementary Fig. 6E). Altogether with these data, ArchT expression within the striatum was specific to the D2-MSNs (Supplementary Table 1).

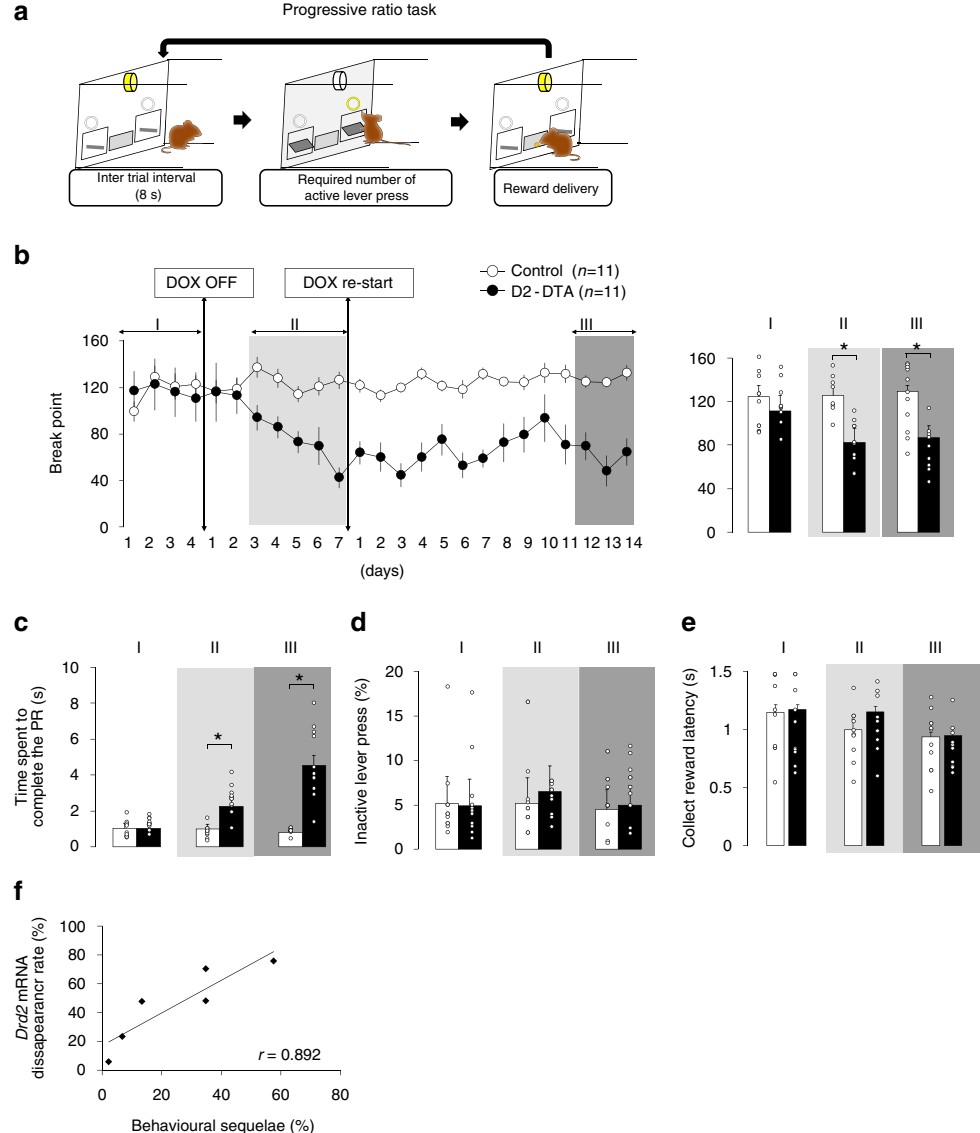

**Figure 5 | Ablation of VLS D2-MSNs produces chronic decreased motivation.** (**a**) Schematic illustration of the PR task sequence. (**b–e**) Temporal changes in behavioural parameters in the PR task. DOX-off for 7 days followed by a restart regimen was used to confine DTA induction within the VLS. The bar graphs show the behavioural parameters averaged over −3 to 0 days before (phase I), 3–7 days after (phase II) DOX termination and 12–14 days after DOX restart (phase III) in the PR task. The black circles represent data from the bigenic group ($n = 11$) and white circles represent data from monogenic controls ($n = 11$). DOX termination decreased break points (**b**, phase II and III) and prolonged time spent to complete the PR (**c**, phase II and III). The DOX-off treatment did not change the number of inactive lever presses (**d**) or collect reward latency (**e**), indicating that associative learning and appetite were spared throughout the experiment. Bars represent the mean and lines represent the s.e.m. *$P < 0.05$, with *post hoc* Student's *t*-test compared with monogenic control. (**f**) The *Drd2* mRNA disappearance rate was correlated with change in break point (behavioural disruption). The behavioural disruption was calculated as: ((break point during DOX restart (days 12 to 14)) in control mice minus break point during DOX restart ((days 12 to 14) in bigenic mice)/ (break point during DOX restart (days 12 to 14) in control mice) × 100).*$P < 0.05$, Pearson's product-moment correlation coefficients.

Optical fibres were implanted into the bilateral VLS in D2-ArchT bigenic mice (Fig. 6a). After re-training, D2-ArchT bigenic mice were illuminated for 2 seconds with yellow or blue light only at the first lever presentation in every trial (Fig. 6d,e). We observed behavioural reductions during the yellow light session in break point ($t_{15} = 6.512$, $P < 0.001$, Fig. 6f), first lever press latency ($t_{15} = 2.912$, $P = 0.011$, Fig. 6g), time spent to complete the PR ($t_{15} = 2.340$, $P = 0.048$, Fig. 6h), but not in collect reward latency ($t_{15} = 0.58$, $P = 0.955$, NS, Fig. 6i) or %inactive lever press ($t_{15} = 1.20$, $P = 0.245$, NS, Fig. 6j) compared with the blue light session. Yellow light-induced behavioural alterations were returned to the pre-stimulus baseline on the following

day (break point: $F_{2,30} = 0.183$, $P = 0.833$, NS, Fig. 6f; first lever press latency: $F_{2,30} = 0.102$, $P = 0.903$, NS, Fig. 6g; time spent to complete the PR: $F_{2,30} = 2.135$, $P = 0.136$, NS, Fig. 6h), indicating that the optogenetic inhibition was transient and reversible.

We further developed an optogenetics-mediated targeted cell ablation technique. This method allowed us to recapitulate the DTA-mediated cell type-specific loss-of-function study. After completion of the acute optogenetic inhibition tests, we applied a 3-h continuous yellow light illumination onto the VLS. Long-term ArchT activation led to cell death, perhaps because of an excess of intracellular alkalization. Such illumination resulted in

the reduction of GFP immunoreactivity below the tip of the fibre, the appearance of single-strand DNA (ssDNA), the decreased number of NeuN positive cells, the apparent loss of *Drd2* mRNA positive medium-size cells (while sparing *Drd1* mRNA positive cells) and the activation of microglial cells, suggesting a D2-MSN-specific ablation (Fig. 6k; Supplementary Fig. 7).

After optogenetics-mediated VLS D2-MSN-targeted ablation, break points were significantly decreased (phase × group interaction: $F_{1, 10} = 3.854$, $P = 0.037$, Fig. 6l) and first lever press latency was prolonged (phase × group interaction: $F_{1,10} = 13.257$, $P = 0.005$, Fig. 6m), effects not seen in controls (break point: $t_{10} = 2.429$, $P = 0.049$; first lever press latency: $t_{10} = 2.628$,

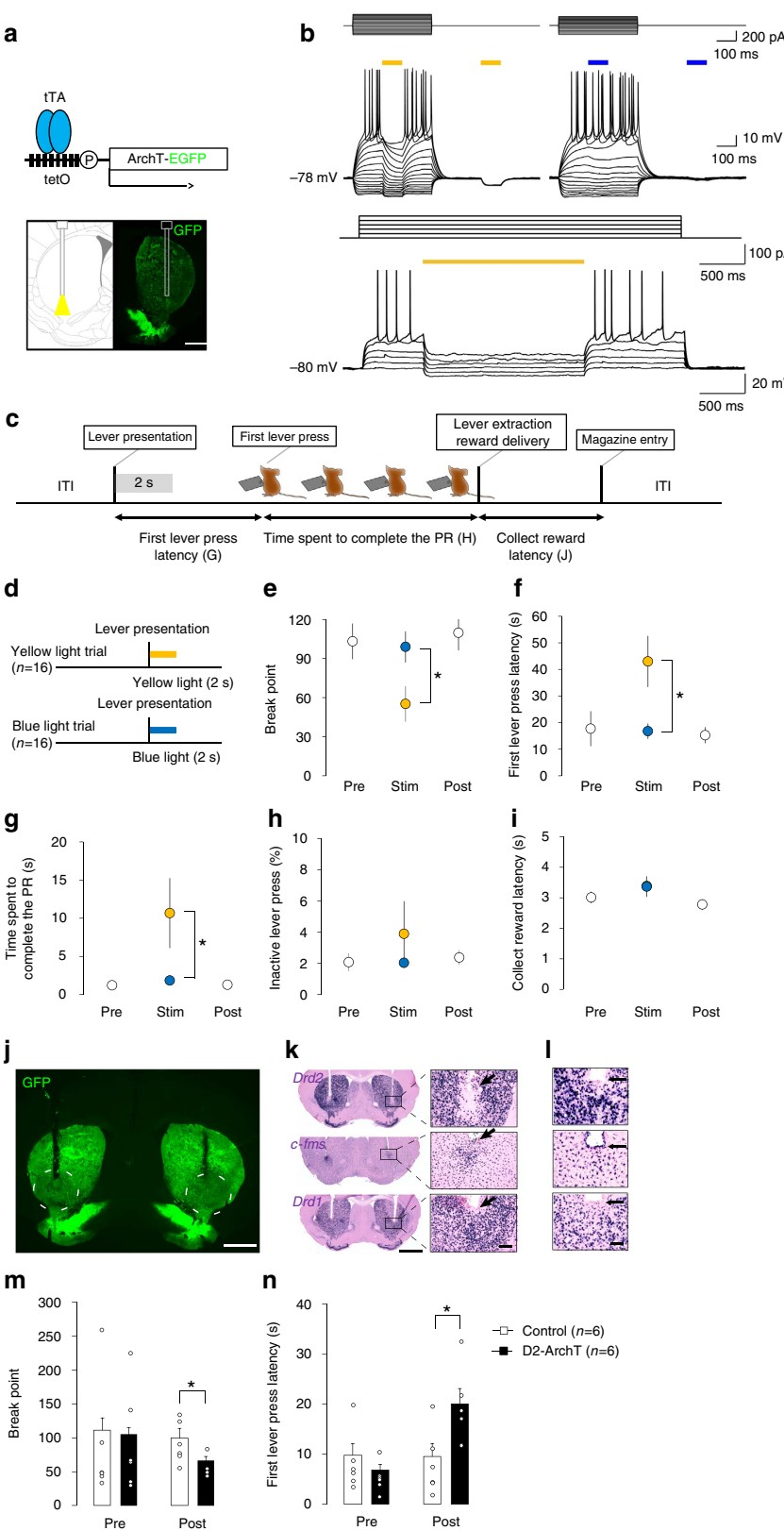

$P = 0.025$, Fig. 6l,m). These data are compatible to results from the DTA-mediated VLS D2-MSN chronic loss-of-function study (Fig. 5b phase III).

## Discussion

We demonstrated that both DTA-mediated cell dysfunction/ablation and optogenetics-mediated inhibition/ablation selectively targeted ventrolateral striatal D2-MSNs (Figs 1 and 6). Two distinct loss-of-function manipulations consistently produced a quantitative reduction of goal-directed behaviours in mice (Figs 4–6). Importantly, reward preference (Figs 4e and 5e, Fig. 6i and Supplementary Fig. 4A), emotional regulation (anxiety-like behaviour (Supplementary Fig. 4B), despair-like behaviour (Supplementary Fig. 4C)), associative learning (Figs 4d and 5d, Fig. 6h) and spontaneous behaviour (Figs 2b and 4f) were not altered by D2-MSN dysfunction, suggesting that D2-MSN dysfunction specifically impairs goal-directed behaviour.

One of the core symptoms of Huntington's disease, which is characterized by the preferential loss of striatal D2-MSNs[13,14], is apathy[4,5]. However, in previous animal studies, ablation of D2-MSNs led to increments in motivated behaviour[26], which suggested that apathy in Huntington's disease may be unrelated to the loss of D2-MSNs. Here we investigate the effect of D2-MSN loss-of-function on motivated behaviour and demonstrate that the striatum contains distinct populations of D2-MSNs with opposing effects on motivated behaviour. Loss of D2-MSNs in the VLS suppressed motivated behaviour toward food (Figs 4b,c and 5b), whereas an expanded loss of D2-MSNs including VMS led to increases in impulsive behaviour that may mask deficits in motivation (Supplementary Fig. 4D,E). Our results identify VLS D2-MSNs as a critical substrate for goal-directed behaviour and suggest this population as a target for treating decreased motivation associated with Huntington's disease and other neurological and psychiatric conditions.

The VLS mainly receives glutamatergic afferents from the IC and projects GABAergic efferents mainly to the lateral (or caudal) VP (refs 18,27). It remains to be elucidated whether this particular cortico-striato-pallidal circuit is involved in motivated behaviour. In humans, medial prefrontal cortical or large frontal lesions are thought to be associated with decreased self-initiated action, a key dimension of apathy[28], suggesting that the IC in particular is not considered a region corresponding to apathy. However, there have been reports implicating IC in information processing during motivated behaviour. For example, activity of the bilateral IC is involved in cost/benefit decision-making[29,30] and is a strong predictor of choosing a low-effort option[31,32]. Thus, it is worth investigating the role of IC-VLS connectivity for a better understanding of apathy and motivation.

Our findings support the role of D2-MSNs in maintaining effort-related motivation but suggest that the role of D2-MSNs in food-seeking and drug-seeking are anatomically dissociable. Theories of drug addiction posit that D2-MSNs negatively control drug-seeking-related motivation. Pharmacogenetic inactivation of D2-MSNs in the nucleus accumbens medial shell, the most medial part of the rostroventral striatum, enhances goal-directed behaviours reinforced by drug but not by food[23]. On the other hand, our data indicate that DTA-mediated dysfunction in the rostrocaudal VLS suppresses food-motivated behaviour while failing to change the incentive properties of addictive drugs (methamphetamine locomotor sensitization and conditioned place preference, Supplementary Fig. 4F,G). This apparent dissociation suggests that the roles of D2-MSNs in drug-seeking and food-seeking are sub-region dependent.

The tetracycline-controllable gene induction system sometimes yields idiosyncratic yet reliable gene induction[33]. In our case, DTA-mRNA induction always initiated in the VLS and expanded concentrically, which led to an advantage in our current work. This unique induction frequently happens if the tetO line is generated by a plasmid transgenic approach, in which tetO cassettes are randomly inserted. Hence this randomness is also a disadvantage because it would be difficult to arbitrarily prepare another tetO-DTA line with different properties. Therefore, it would be difficult to address a DTA-mediated loss-of-function effect in other parts of the striatum using this same method, which is distinct from systems exploiting diphtheria toxin receptor expression and focal DT injection[17,26] or immunotoxin-mediated targeting[34].

DTA-mediated cell death-based evaluation of circuitry presents caveats. Cell death inevitably coincides with glial activation (Fig. 1e,f). Glial activation could have indirect consequences that may alter circuit function nonspecifically. The optogenetic approach using ArchT expression would be a desirable means to address the above caveat. We employed acute optogenetic inhibition (Fig. 6) that did not induce glial activation and observed a behavioural reduction in goal-directed behaviour. We also employed DOX-off and restart regimen (Fig. 5) in which only a subset of D2-MSNs was ablated and the resultant glial activation was alleviated (Supplementary Fig. 8). This experiment, too, produced a reduction in goal-directed behaviour. Although these additional experiments strongly support a specific role of D2-MSNs, it is impossible to exclude non-cell autonomous effects related to D2-MSN degeneration. Because neurodegenerative disorders and cerebral vascular disease always include nonspecific effects accompanying neuronal damage, the possibility of nonspecific effects may strengthen the construct validity of our animal model of neurodegeneration rather than weaken it.

**Figure 6 | Optogenetics-mediated inhibition/ablation of VLS D2-MSNs induces reductions in goal-directed behaviour.** (**a**) Drd2-tTA-dependent ArchT expression (upper) and illumination of bilateral VLS D2-MSNs (lower). Scale = 1 mm. (**b**) ArchT-expressing MSN response to current injection and illumination during whole cell voltage recording. Yellow, but not blue, light illumination suppressed action potential generation (upper panel). Two seconds-yellow light illumination suppressed action potential generation without inducing rebound excitation (lower panel). (**c**) Timing of illumination in the PR task. Yellow or blue light was applied immediately after every lever presentation for 2 s (grey shade). ITI: inter-trial interval. (**d**) The experimental schedule of optogenetic inhibition (eight D2-ArchT bigenic mice were used and 16 sessions were conducted). (**e–i**) The effects of acute optogenetic inhibition of VLS D2-MSN on behavioural parameters in the PR task. The yellow-light illumination decreased break point (**e**) and prolonged first lever press latency (**f**) and time spent to complete the PR (**g**). These behavioural changes were restored to the pre-baseline levels on the following day. Yellow-light illumination did not change the collect reward latency (**h**) or %inactive lever press (**i**). (**j–l**) Continuous 3-h illumination induced a cell type-specific ablation in the VLS. GFP florescence was attenuated in the VLS (**j**) Scale, 1 mm. In the D2-ArchT bigenic mice (**k**), Drd2 mRNA labelling disappeared (upper) and c-fms-positive microglia were activated (centre) while Drd1 mRNA expression did not change (lower) in the area below the tip of the optic fibre (arrow). Scale, 50 μm. In the monogenic controls (**l**), no apparent changes were observed in Drd2 (upper), c-fms (centre), or Drd1 (lower) mRNA expressions. Scale, 50 μm. (**m,n**) The effects of optogenetic ablation of VLS D2-MSNs on behavioural parameters in the PR task ($n = 6$ for each group). The continuous 3-h yellow-light illumination decreased break point (**m**) and prolonged first lever press latency (**n**). *$P < 0.05$, Paired t-test compared with the monogenic control.

Optogenetic inhibition of the VLS D2-MSNs phenocopied the region-specific manipulation by DTA. The behavioural alterations produced by optogenetic inhibition were similar to those associated with *DTA* mRNA expression without cell death (Fig. 5b, phase II), strengthening the notion that viable DTA-expressing cells were hypoactive (Fig. 3). Optogenetic ablation is a new tool to achieve targeted cell ablation (Fig. 6j,k); optogenetic inhibition confirmed the functional readout of acute silencing of specific circuitry, followed by optogenetic ablation. This combination enables us to understand the effects of transient and permanent loss-of-function of identical cell types within the same animal. Another interesting point from our optogenetics study was the timing of optogenetic silencing during the task (Fig. 6d). Two seconds of stimulation, time-locked to the first lever presentation in every trial, was sufficient to decrease motivated behaviour (Fig. 6e–g), leading us to hypothesize that VLS D2-MSNs firing at this time point may represent the trigger of self-generated action. Further efforts including the analysis of D2-MSN activity during the task will be required.

The impaired goal-directed motivation observed in our animal model may provide insight into the pathomechisms of apathy, which is a common feature of neurodegenerative disease. Apathy is defined clinically as a lack of motivation[9,10]. Historically, clinical identification of apathy has been highly subjective, owing to the variety of psychological conceptions of motivation. To improve the objectivity of apathy diagnosis, Levy and Dubois recently proposed that apathy should be defined as a behavioural syndrome consisting of a quantitative reduction of self-generated and voluntary goal-directed behaviours[28]. The widespread acceptance of this definition[35,36] has enhanced the consistency of apathy diagnosis and has made the construct more amenable to modelling in animals. Our animal models with targeted ablation of VLS D2-MSNs appear to fulfil the operational definition of Levy and Dubois. Whether VLS D2-MSN degeneration reproduces all the features of apathy, as commonly defined, is unknown and may be difficult to address in animals. Nevertheless, our studies are a first step towards establishing model animals that can be used in translational research towards new treatments for decreased motivation in neurodegenerative, traumatic or cerebrovascular diseases.

## Methods

**Ethical statement.** All animal procedures were conducted in accordance with the National Institutes of Health Guide for the Care and Use of Laboratory Animals and approved by the Animal Research Committee of the School of Medicine, Keio University.

**Generation of *Drd2*-tTA BAC transgenic mice.** A cassette containing mammalianized tetracycline transcriptional activator (tTA)—SV40 polyadenylation signal was inserted into the translation initiation site of the dopamine receptor type 2 (*Drd2*) gene in mouse bacterial artificial chromosome (BAC) DNA (clone RP23-161H15). Modified linearized BAC DNA was injected into fertilized eggs from CBA/C57BL6 mice. We generated only one founder.

**Mouse maintenance and genotyping.** All mice (C57BL/6J,129SvEvTac and CBA mixed background, male) were maintained with 12:12-h light/dark cycle (lights on at 8:00 h) and the behavioural experiments were conducted during the light phase. tetO-diphtheria toxin A subunit (tetO-DTA) mouse was obtained from Jackson Laboratory and tetO-ArchT-EGFP mouse was from RIKEN Bioresource Center.

The following sets of primers were used for genotyping: D2R-582U (5′- GCG TTTGACTAAGTTGCCAAGCTG-3′) and mtTA24L (5′-CGGAGTTGATCA CCTTGGACTTGT-3′) was used for *Drd2*-tTA mice and yielded a 610 bp band; tetOup (5′-AGCAGAGCTCGTTTAGTGAACCGT-3′) and DTAlow (5′-GGC ATT ATC CAC TTT TAG TGC-3′) was used for tetO-DTA mice and yielded a 370 bp band; YFPup (5′- CATGAAGCAGCACGACTTCTT-3′) and YFPlow (5′- TTCTTACTTGTACAGCTCGTCCA-3′) was used for tetO-ArchT-EGFP and

yielded a 470 bp band. Wild type mice did not yield any band. From the start of the breeding, *Drd2*-tTA::tetO-DTA bigenic mice were fed with doxycycline (DOX)-containing chow (100 mg DOX kg⁻¹ chow, CLEA Japan, Tokyo, Japan) to avoid DTA induction. *Drd2*-tTA::tetO-ArchT-EGFP bigenic mice were fed with normal chow (CE-2, CLEA).

**Immunohistochemistry.** Mice were deeply anesthetized and perfused with 4% paraformaldehyde in 0.1 M phosphate buffer (PB). Brains were removed from the skull and postfixed overnight. After overnight cryoprotection with 20% sucrose/PB, brains were frozen and cut at 25 μm thickness. Sections were mounted on silane-coated glass slides (Matsunami Glass, Tokyo, Japan). For the antigen retrieval, sections were heated at 98 °C in 10 mM citrate buffer (pH 6.0) for 40 min. Sections were incubated with the primary antibodies overnight at room temperature. The following antibodies were used: anti-dopamine receptor type 1 (Drd1) (D1R-Go-Af100, 1:1,000, goat polyclonal, Frontier Institute Co. Ltd, Sapporo, Japan), anti-Drd2 (D2R-GP-Af500, 1:1,000, guinea pig polyclonal, Frontier Institute), anti-green fluorescent protein (GFP) (GFP-Rb-Af2020, 1:1,000, rabbit polyclonal, Frontier Institute), anti-single-strand DNA (#18731, 1:1,000, rabbit polyclonal, IBL, Fujioka, Japan), anti-glial fibrillary acidic protein (GFAP) (G3893, 1:1,000, mouse monoclonal, clone GA5, Sigma-Aldrich Japan, Tokyo, Japan) and anti-ionized calcium binding adaptor molecule 1 (Iba1) (019-19741, 1:500, rabbit polyclonal, Wako Chemical, Odawara, Japan). For fluorescence microscopy, sections were treated with solo or a mixture of species-specific secondary antibodies conjugated to Alexa Fluor 488, 555 or 647 (1:1,000, Invitrogen, Grand Island, NY) for 2 h at room temperature. Fluorescent images were obtained with a confocal microscope (FV1000; Olympus, Tokyo, Japan). For light microscopy, sections were treated with species-specific biotinylated secondary antibodies (1:250, Vector Laboratories, Burlingame, CA) for 90 min, an avidin-biotin complex (Elite ABC kit, Vector Laboratories) for 30 min, and 3,3′-Diaminobenzidine (ImmPACT DAB, Vector Laboratories). Images were obtained with an inverted light microscope (BZ-X710, Keyence, Osaka, Japan).

**Colorimetric In situ hybridization (ISH).** Cryosections from fixed brains were treated with proteinase K (40 μg ml⁻¹; Merck). After they were washed and acetylated, sections were incubated with a digoxigenin (DIG)-labelled complementary RNA (cRNA) probe. After the sections were washed in buffers with serial differences in stringency, they were incubated with an alkaline phosphatase-conjugated anti-DIG antibody (11093274910, 1:5,000; Roche, Japan). The cRNA probes were visualized with freshly prepared colorimetric substrate (NBT/BCIP; Roche, Japan). Nuclear fast red (Vector Labs., Burlingame, CA) was used for counterstaining[37]. We employed probes for *DTA, Drd1, Drd2, tTA*, choline acetyltransferase (*ChAT*), dopamine transporter (*DAT*) and *c-fms*. Plasmid information is available on request.

**Double fluorescent ISH.** Sections were hybridized with fluorescein isothiocyanate (FITC)-labelled *ChAT* cRNA probe and digoxigenin (DIG)-labelled *Drd2* or *tTA* probe. After a stringent wash, sections were incubated with peroxidase-conjugated anti-DIG antibody (11207733910, 1:1000; Roche) and labelled with Cy3 by using the tyramide signal amplification (TSA) system (Perkin-Elmer, Waltham, Massachusetts). Followed by quenching with 1% H₂O₂, sections were incubated with peroxidase-conjugated anti-FITC antibody (11426346910, 1:1500; Roche) and labelled with FITC by the TSA system.

**Quantification of *Drd2* mRNA disappearance rate after DOX withdrawal.** Ten coronal sections from Bregma +1.8 mm to −0.4 mm with 200 μm interval were mounted on one glass slide. ISH of *Drd1* and *Drd2* was conducted with serial sections and the digital colour images were captured at 3048 dpi ('ISH image': Supplementary Fig. 2A). *R* value was isolated from RGB colour data ('*R* value image (grayscale)'). Binary threshold was determined by being under the mean *R* value at dorsolateral striatum and 'Binary image' was obtained. The number of positive pixels was counted in both Drd1 and Drd2 slides. *Drd2* mRNA disappearance value per animal was calculated as ((Drd1 pixels − Drd2 pixels)/ Drd1 pixels) and averaged (n = 5). Subtraction of averaged *Drd2* mRNA disappearance value of DOX-on control from that of DOX-off treatment was defined as *Drd2* mRNA disappearance rate.

**Determination of averaged *Drd2* mRNA disappearance area.** The colour images of Drd1- or Drd2-labelled unilateral striatum at Bregma +1.2 mm, +0.4 mm and −0.4 mm were captured at the aspect ratio 1.5 ('ISH image': Supplementary Fig. 2A). After *R* value extraction ('*R* value image (grayscale)'), the resolution was reduced to 600 pixel/image (20 × 30 grids) ('*R* value image (Low-resolution)'). The value of each pixel was imported to an Excel worksheet grid and was adjusted from 0 to 100. ('Text data of adjusted *R* value'). The value of Drd2 was subtracted from that of Drd1 ('Subtracted value (Drd1−Drd2)'). Subtracted values were averaged in each grid (n = 5 animals). In each grid, the proper value was obtained by the normalization: averaged valueDOX-off was subtracted from averaged valueDOX-on. The grey-scale image was reconstructed from

the information of the grid position and its proper value (Supplementary Fig. 2C). From the minimum proper value to the value 50, a linear grayscale was adapted and over 50 values indicated darkened colour.

**Multichannel extracellular recording.** Mice were anesthetized with sevoflurane (1.5–3%, 0.25 ml min$^{-1}$) and were fixed to a stereotaxic apparatus (SM-15, Narishige Scientific Instrument, Tokyo, Japan). Their body temperature was maintained at 37 ± 0.5 °C using a heating pad (FHC-MO, Muromachi Kikai, Tokyo, Japan) during surgical and recording procedures. For stimulating the insular cortex, a glass-coated tungsten wire (250 μm diameter, 0.5 MΩ, Alpha Omega, GA, USA) was inserted at AP + 2.0 mm, ML + 2.0 mm and depth of 2.0 mm. Electrical stimuli consisted of monopolar pulses of 200 μs width and 200 μA intensity delivered at a frequency of 0.5 Hz. After electrophysiological recordings, the tip of the electrode was marked by passing 20 μA for 30 s in both polarities. For multi-channel extracellular recordings in the ventral pallidum, a silicon probe (linear 16-electrode array, 100 μm interval, 177 μm$^2$ recording site area, 15 μm thickness, NeuroNexus Technologies, MI, USA) was set at 0.1 mm posterior to the Bregma (AP − 0.1 mm), 2.0 mm lateral from the midline to the left (ML + 2.0 mm). The probe was coated with DiI, a lipophilic fluorescent dye (D-282, Invitrogen Life Technologies Japan, Tokyo, Japan; 80 mg ml$^{-1}$ in a 50:50 methanol:acetone mixture).

Electrophysiological signals were recorded at 24.42 kHz with a band-pass filter (0.3 Hz–5 kHz) (TDT RZ-2, Tucker-Davis Technologies, FL). A silicon probe was inserted vertically into the brain through the dura mater using a microdrive. The ventral pallidum is located just beneath the globus pallidus externa (depth 3.0–4.0 mm), a region where high frequency unit activities (80–100 counts per s) were detected. We first targeted the lateral portion of the globus pallidus externa, and then reached the lateral portion of the VP, where moderate frequency unit activities (40–60 counts per s) were detected (depth 4.0–4.5 mm). In both DOX-on and -off periods, baseline firing of VP were comparable as previous immunotoxin-mediated cell ablation study reported[34]. All brains were collected to determine the positions of stimulation and recording. Each response pattern was randomly obtained in the ventral pallidum in DOX on and off regimens.

**Data analysis of *in vivo* electrophysiology.** Local field potentials (LFPs) were analysed using custom-written programs in MATLAB. As a preprocess, voltage transients evoked by electrical stimulation (i.e., artifact) were removed by a template subtraction method, in which a template was subtracted from original signals. The template was constructed by averaging stimulus-triggered LFPs (10-ms-long after stimulus onset) across all stimulations ($N = 100$ in a session). Since the amplitudes and baselines of evoked voltage transients are validated stimulus-to-stimulus, the template was fitted to each stimulus-triggered LFPs by the steepest descent method.

For analysis of multi-unit spikes (MUs), the LFPs were first Butterworth-filtered (300-Hz high pass with zero-phase filtering). MUs were then detected when the negative peak of the filtered LFPs exceeded a threshold 3σ, where σ = median (absolute value of the filtered LFP/0.6745) (ref. 38). The waveform of the MUs was extracted for 0.5 ms before and for 1.0 ms after the detection of the MUs. Implausible MUs, which showed (1) low correlation ($< 0.7$) in waveform with MUs of baseline (for 300 ms before the stimulation) and (2) larger absolute value of positive voltage compared with that of negative voltage, were excluded for subsequent analysis.

The neuronal responses to IC electrical stimulation were assessed by constructing peristimulus time histograms (PSTHs; bin width of 1 ms) using 100 stimulation trials. The mean value and s.d. of the firing rate during the 100 ms preceding the onset of stimulation were calculated from a PSTH and were considered the baseline discharge rate. Changes in firing activity in response to cortical stimulation (that is, excitation and inhibition) were judged to be significant if the firing rate during at least two consecutive bins (2 ms) reached the statistical level of $P < 0.05$ (one-tailed $t$-test)[34]. The methods to determine the latency and the duration of each response were described previously[34].

**Open Field Test (OFT).** All D2-DTA bigenic animals were fed with DOX chow by 6 weeks of age. Normal chow feeding started at 6 weeks of age and lasted for 10, 14 or 28 days, and then DOX chow feeding restarted and lasted until 11 weeks of age. Bigenic littermate controls were fed with DOX chow throughout. OFT was conducted at 11 weeks of age. OFT was performed in a 36 × 36 × 26 cm$^3$ white-coloured box for 30 min. The brightness of the field was 50 lux. The behaviours were recorded by a camera (2 MP Webcam C600, Logicool, Tokyo, Japan). Moving distance was calculated by a self-produced MATLAB program.

**Three-choice serial reaction time task (3-CSRTT).** Six D2-DTA bigenic mice and six monogenic mice (*Drd2*-tTA, n = 3; tetO-DTA, n = 3) were subjected to the 3-CSRTT. They were housed individually and received food restriction. Thereafter, their body weights were maintained at 85% of weights under free-feeding conditions. They received 2.3–3.0 g of DOX chow per day, and no *DTA* mRNA induction was observed.

Aluminium operant chambers measuring W22 × D26 × H18 cm (Med Associates Inc., St. Albans, VT, USA) were used. The curved rear wall of each chamber contained nine holes. Each hole had an infrared photocell beam for

detection of nose-poke responses and a 2.8 W bulb at its rear. The side holes were sealed so that only the three centrally positioned holes were accessible. A food magazine was located on the opposite wall of the chamber, and a house light was located at the top of this wall. The apparatus was controlled by a computer program written in the MED-PC language (Med Associates Inc.).

We used the same training procedure as previously described with some modifications[21]. In the beginning of the training phase, all hole lights were illuminated for 30 s. In the second phase, one hole light was illuminated randomly for 30 s. In the subsequent phases, the stimulus duration was decreased in a stepwise manner as the training progressed (stimulus duration 15, 10, 7, 5, 4, 3, 2, 1.5 and 1 s). After completion of the training, the stimulus duration was fixed at 1 s regardless of performance. We set the criteria for determining stable performance as follows: the variability of premature response (no.), accuracy (%) and omission (%) in the last three sessions were <20%, 10% and 20% respectively. Training was conducted for one session per day.

The task sequence used in the 3-CSRTT was the same as our previous study using the 3-CSRTT with some modifications[21]. Each session began with turning on the house light and food magazine light. The first trial began when a mouse entered the magazine, terminating the house light and magazine light. After a 5-s delay (waiting time) one of three hole lights was briefly illuminated (stimulus duration) in a pseudo-random order so that the mouse could not predict which hole would be illuminated. Nose poking during the waiting time was recorded as a premature response. Nose poking into the lit hole while it was illuminated or within 5 s of limited hold was recorded as a correct response, and resulted in illumination of the house light and delivery of a palatable food pellet (20 mg each, dustless precision pellets, Bio-serv, Frenchtown, NJ, USA) with illumination of the magazine light. Once the mouse entered the magazine to receive his food reward, the magazine light was again terminated (initiation of inter-trial interval). After a 0.5-s delay, the magazine light and house light flashed for 0.5 s and then these lights were continuously illuminated. Another nose poke into the magazine resulted in the termination of magazine and house lights, indicating the start of a new trial. Nose poking into the non-lit hole was recorded as an incorrect response. When the animal failed to nose poke within the limited hold time (5 s), it was recorded as an omission. Additional nose poking into any of the three ports before food collection was recorded as a perseverative response.

Premature responses, incorrect responses, omissions and perseverative responses resulted in a 5 s time-out period during which the house light was illuminated. After the time-out period, the magazine light and house light flashed for 0.5 s and then these lights were continuously illuminated. Nosepoke into the magazine was required to start the next trial. Only in cases of premature response, nosepoke into the magazine after the time-out period restarted the same trial.

Simultaneously, correct response latency (the mean time between stimulus onset and nose poke into the correct hole) and reward latency (the mean time between reward delivery and nose poke into the food magazine) was recorded and regarded as attentional/motor function and motivation/appetite, respectively. The number of total trials was regarded as an index of motivation. Accuracy (correct responses/(correct and incorrect responses) × 100) was calculated and regarded as an index of attentional function. % omission (the number of omissions/total trial × 100) was calculated and regarded as an index of sustained motivation/attention. % premature response (the number of premature responses/total trial × 100) was calculated and regarded as an index of impulsive action. % perseverative response (the number of perseverative responses/total trial × 100) was calculated and regarded as an index of compulsive behaviour.

The behavioural data were analysed in three phases: the phase before loss-of-function manipulation is classified as Phase I; the phase from appearance of *DTA* mRNA in the VLS (DOX off days 3) to the appearance of cell death (DOX off days 10) as is classified as Phase II; and the phase after the cell death expansion to the whole VS (DOX off days 14) is classified as Phase III.

**Progressive ratio (PR) task.** Six D2-DTA bigenic mice and six monogenic mice (*Drd2*-tTA, n = 3; tetO-DTA, $n = 3$) were subjected to the PR task. These mice did not receive any other operant training. The housing and food restriction conditions were comparable to those applied in the 3-CSRTT.

The same apparatus used in the 3-CSRTT with some modifications was used in the PR task. The nine holes were sealed with aluminium plates. Two retractable levers were equipped on either side of a food magazine.

Mice are initially trained to press the lever on a fixed ratio (FR)-1 reinforcement schedule whereby a single lever press elicits the delivery of a food pellet to the magazine. A trial was started with the house light off and two levers presented. Only one lever is designated as 'active' (triggering delivery of food reward) and the allocation of right and left levers was counterbalanced between mice. After the food delivery, 8 s of inter-trial interval was added, during which levers were retracted, followed by automatic starting of the next trial. The inter-trial interval allows time for mice to consume the food pellet. Following two successive sessions of obtaining ≥50 pellets, the schedule was increased to FR-2 in which two active lever presses triggered the delivery of the food pellet. Training on the FR-2 schedule lasted three days. Then, the schedule was increased to FR-3 and lasted three more days. Each FR training session lasted 1 h or when 100 pellets had been delivered.

The mice are then trained in the PR schedule of reinforcement[24]. The response ratio schedule during PR testing can be calculated according to the following formula:

$$= \left[ 5e^{(R \times 0.2)} \right] - 5$$

where $R$ is equal to the number of food rewards already earned plus 1 (that is, the next reinforcer). Thus, the number of responses required to earn a food reward followed the order: 1, 2, 4, 6, 9, 12, 15, 20, 25, 32, 40, 50, 62, 77, 95, and so on. The final ratio completed represented the break point. A PR session lasted a maximum of 1 h. Failure to press the lever in any 5 min period resulted in termination of the session. Performance on the PR schedule of reinforcement was considered stable when the number of rewards earned in a session deviated by ≤10% for at least 3 consecutive days. The break point was recognized as an index of instrumental motivation. The time spent to complete the PR, the mean time from the first active lever press to achieving the required number of active lever press, was recognized as an index of instrumental motivation. %inactive lever press (inactive lever press/(inactive and active lever press) × 100) was calculated and was recognized as an index of associative learning. The latency to collect reward was recognized as an index of appetite.

The behavioural sequelae were calculated as: (break point during DOX restart (days 12 to 14)) in control mice minus break point during DOX restart (days 12 to 14) in bigenic mice)/(break point during DOX restart (days 12 to 14) in control mice) × 100).

**Elevated plus maze (EPM).** Nine D2-DTA bigenic and nine monogenbic control animals were fed with DOX chow by 7 weeks of age. Normal chow feeding started at 7 weeks of age and lasted for 7 days. The test apparatus consisted of two of both open and closed arms ($25 \times 5\,cm^2$) that extended from a central platform ($5 \times 5\,cm^2$). Closed arms were surrounded by 40 cm high-walls. The maze was elevated 40 cm above the floor, and the room lights were turned on (>100 lux) during testing. Mice typically avoid the open arms because they innately dislike open space, and anxiolytic agents increase the time spent in open arms[39]. That is, a decrease in the time spent in the open arms indicates an increase in anxiety. In addition, the distance travelled in the maze was used to quantify locomotor activity. The test was initiated by placing the mouse on a central platform facing an open arm, and the recording was initiated once the mouse entered a closed arm. If a mouse failed to enter a closed arm after 1 min, data from that mouse were excluded from the analysis (one mouse were excluded). The mouse was recorded by a WEB camera over a 5 min period; the recorded data were analysed automatically using a software package (Any-Maze, Stoelting Co., USA).

**Forced swimming test (FST).** Nine D2-DTA bigenic and nine monogenbic control animals received FST after the testing of EPM (>2-h interval was taken). Mice were placed in a cylinder glass (diameter, 23 cm; height, 30 cm; Iwaki, Japan) containing water at a temperature of 24 °C ± 1 °C and a depth of 12 cm; mice could not escape nor touch the bottom of the cylinder. The swimming test lasted 6 min and behaviours were video-recorded and scored later by an experimenter who was blind to the treatments. Only the last 4 min of behaviours were analysed, and they were categorized as follows: Immobility, the mouse is completely still in the water, except for isolated movements to right itself; climbing, movement of all four legs with the body aligned vertically in the water[40].

**Food preference test and food consumption test.** Forty D2-DTA bigenic animals received food preference and consumption tests. They were divided into five groups; DOX-on, -off 5, -off 7, off-10, off-14 days ($n = 8$, each). Food-restricted mice (home cage chow was restricted to 2.0 g per day) were allowed ad libitum access to both normal chow and palatable sucrose pellets (the same product used in operant tasks) for 1 h in a new cage. About 4 h after the completion of the food preference test, mice received food consumption test. The mice were allowed ad libitum only access to palatable sucrose pellets for 1 h in a new cage. Each test was conducted three consecutive days and data of the last day was used in statistical analysis.

**Methamphetamine (METH) sensitization.** Twenty three D2-DTA bigenic animals (DOX-on, $n = 11$; DOX-off day 10, $n = 12$) received this test. The DOX-off schedule was the same as described in the OFT procedure. Locomotor activity in the OFT box was measured for 25 min immediately after intraperitoneal injection of METH (1 mg kg$^{-1}$, Dainippon Sumitomo Pharma, City, Japan). METH was administered for 5 consecutive days. Travel distance was calculated as described in the OFT procedure.

**METH conditioned place preference.** Twenty D2-DTA bigenic animals (DOX-on, $n = 10$; DOX-off day 10, $n = 10$) received this test. The DOX-off schedule was the same as described in the OFT procedure. For the conditioned place preference task, the apparatus consisted of two compartments: a white Plexiglas box and a black Plexiglas box (both $15 \times 17 \times 16\,cm^3$ high). The floors of the white and black boxes were covered with white plastic mesh and black frosting Plexiglas, respectively. Each box could be divided by a sliding door ($15 \times 19\,cm^3$ high). The place conditioning paradigm was performed as an abbreviated version of a

previously established procedure[41]. On the first day, the preconditioning test was performed. The sliding door was opened, and the mouse was allowed to move freely between both boxes for 15 min (twice per day). At the second trial, the time that the mouse spent in the black and white boxes were recorded by using a web camera (DC-NCR300U, Hanwha Q CELLS Japan, Tokyo, Japan) and measured by an observer. The box in which the mouse spent the most time was referred to as the 'preferred side' and the other box as the 'nonpreferred side'. Conditioning was performed on the subsequent 3 successive days. Mice were given METH (1.0 mg kg$^{-1}$) or saline (10 ml kg$^{-1}$) injections and put into the apparatus with the sliding door closed immediately after the injection. That is, a mouse was intraperitoneally given saline and placed in its preferred side for 15 min in the morning. In the afternoon of the same day, the mouse was given METH and placed in its nonpreferred side for 15 min. In the post-conditioning test, the sliding door was opened, and the time that the mouse spent in the black and white boxes was recorded for 15 min. Place conditioning behaviour was expressed by Post-Pre, which was calculated as: ((postvalue) − (prevalue)), where postvalue and prevalue were the difference in time spent in the drug conditioning and the saline conditioning sites in the postconditioning and preconditioning tests, respectively.

**Optogenetics-mediated inhibition/ablation.** Approximately 8 D2-ArchT bigenic mice were subjected to the PR task under the influence of optogenetic inhibition. After completing PR training, mice were anesthetized with a ketamine-xylazine mixture (100 mg kg$^{-1}$ and 10 mg kg$^{-1}$, respectively, i.p.) and fixed in a stereotaxic frame (Narishige, Tokyo, Japan). Animals were bilaterally inserted with a 200 μm core diameter optic guide cannulae (Thorlabs, NJ) into the VLS (bregma + 0.4 mm, lateral ± 2.0 mm, depth − 4.0 mm from the skull surface). After surgery, the mice were housed individually and allowed a 4-day recovery period before retraining. Mice were then retrained on the PR task for at least one week until their performance stabilized for three consecutive sessions. Before each experiment, optical fibres (Doric Lenses, Quebec, Canada) were inserted bilaterally through the guide cannulae. Yellow (575 nm, 3 mW each) or blue (475 nm, 2 mW each) light was generated by a SPECTRA 2-LCR-XA light engine (Lumencor, OR). We did not use green light, which is optimal to activate ArchT, because of our limited experimental setup. Each experimental timeline set consisted of consecutive 3-day sessions. Illumination sessions (Day2 (Stim)) were flanked by no illumination sessions (Day1 (Pre) and Day3 (Post)). Two sessions (yellow or blue, counter balanced) were conducted on Stim and one session was conducted on Pre and Post. Sixteen sets were conducted. In the optogenetic experiments, inter-trial intervals were prolonged to 30 s and the first lever press latency (the mean time from lever presentation to the first active lever press) was calculated as another index of instrumental motivation.

Approximately 6 D2-ArchT bigenic and six D2-tTA monogenic mice received opto-ablation manipulation of the VLS. After completion of the acute optogenetic inhibition tests, we applied a 3-h continuous yellow light illumination onto the VLS. Long-term ArchT activation may cause an excess of intracellular alkalization, leading to cell death. We conducted the PR task 3–1 days before (Pre) and 5–7 days after (Post) the application of optogenetic ablation.

**Patch clamp.** Mice (10 weeks of age) were anesthetized with isoflurane and decapitated. Within 30 s of decapitation, the brain was removed and placed into ice-cold oxygenated (95% O2/5% CO2) cutting solution containing (in mM) 93 N-Methyl-D-glucamine-Cl, 30 NaHCO3, 20 HEPES, 2.5 KCl, 10 MgSO4, 0.5 CaCl2, 1.2 NaH2PO4, 25 glucose, 5 sodium ascorbate, 3 sodium pyruvate and 2 thiourea (pH 7.4). Approximately 350 μm-thick sagittal striatal slices were cut on a microslicer (LinearSlicer Pro 7, Dosaka, Kyoto, Japan) in the ice-cold cutting solution. The slices were incubated for 15 min in the 32 ± 1 °C cutting solution. The slices were then transferred into 22 ± 1 °C normal artificial cerebrospinal fluid (aCSF) containing (in mM) 124 NaCl, 26 NaHCO3, 3 KCl, 2 MgCl2, 2 CaCl2, 1.25 NaH2PO4 and 11 glucose (pH 7.4).

Whole-cell recording pipettes were pulled from borosilicate glass (6–8 MΩ). The pipettes were filled with (mM) 120 K-gluconate, 4 NaCl, 40 HEPES, 2 MgATP and 0.3 Na3GTP, pH 7.2 (adjusted with KOH). Membrane potentials were recorded with a CEZ-2400 amplifier (Nihon Kohden, Tokyo, Japan) and an A-D/D-A converter (20 kHz per channel, 16 bit; USB-6259BNC, National Instruments, Austin, USA). Ventral striatum MSNs were illuminated and visualized using an upright microscope (AxioSkop 2FS, Zeiss, Jena, Germany) with a × 40 water-immersion objective lens. Throughout the recordings, the slices were perfused with 31 ± 1 °C aCSF at ∼2 ml min$^{-1}$.

Amber and blue LED light sources (Luxeon Rebel LXM2-PL01 for amber and LXML-PB01-0023 for blue, Lumileds, San Jose, USA) provided coloured photostimulation through an epi-illumination port with a 605 nm dichroic mirror. LED light pulse intensity and timing regulation was achieved by custom Matlab software, which was used to control a custom-designed current regulator through the A-D/D-A converter.

**Statistical analyses.** Sample sizes were determined on the basis of pilot experiments and previous experience from similar experiments. To examine whether the data had the same variances, we first analysed them by F-test. As all the data were determined to be normally distributed, parametric statistics were used throughout.

Behavioural, histological and electrophysiological data were analysed by Student's $t$-test, $\chi^2$ test or repeated measures ANOVA with Bonferroni *post hoc* test using SPSS software version 20 (IBM, Armonk, New York, USA). All the $t$-tests were performed as two-tailed $t$-tests. The statistical test used for each experiment is stated in the text and/or corresponding figure legend.

**Data availability.** The data that support the findings of this study are available from the corresponding author on reasonable request.

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

## Acknowledgements

This work was supported by Grant for Research Fellow of the Japan Society for the Promotion of Science (2640100) to I.T-K, Schizophrenia Clinical and Basic Research (SCBR) grant to H.T., Grant-in-Aid for Brain Mapping by Integrated Neurotechnologies for Disease Studies (Brain/MINDS) from the Ministry of Education, Culture, Sports, Science, and Technology of Japan (MEXT) to H.O., N.T. and K.F.T., Grant-in-Aid for Scientific Research on Innovative Area 'Microendophenotype' (25116523) and 'Oscillology' (16H01621) from the MEXT to K.F.T., 'Adaptive Circuit Shift' (15H01458) from the MEXT to H.S. Grant-in-Aid for Scientific Research (B) from MEXT to K.F.T. (15H03123), Takeda Science Foundation to K.F.T.

## Author contributions

I.T.-K., H.T., M.X., R.Y., H.O., M.U., and M.W. conducted histological experiments, I.T.-K., H.T., Y.B., and M.M. conducted behavioural experiments, K.Y., O.H., H.N., N.T., and H.S. conducted electrophysiological experiments, K.F.T. designed the experiments, I.T.-K., M.R.D. and K.F.T. wrote the paper.

## Additional information

**Competing financial interests:** The authors declare no competing financial interests.

