## [Peer Review File · Nature Communications]

Reviewers' comments:

Reviewer #1 (Remarks to the Author):

The study by Tsutsui-Kimura investigates the role of D2-MSN on progressive ratio responding and other behaviors controlled by striatal circuits. The authors performed targeted ablation of D2-MSN using a transgenic mouse model that allows for temporal control (using DOX-diphtheria toxin system) and shows region specificity for the ventral striatum (nucleus accumbens) and dorsomedial striatum. The study found a reduction in breakpoint during progressive responding for food reward and a reduction in omission in 3-choice serial reaction time task. No changes in anxiety-like behaviors or forced swim test.

The study takes full advantage of the possibilities of this technology for cell-specific ablation by performing the elegant experiments described in Fig. 5 in which DOX treatment is temporarily suspended for 7 days and then restarted to prevent the continued spread of the ablation. Importantly, a separate set experiments uses an independent approach (optogenetic inhibition and ArchT-mediated ablation) to manipulate the activity of D2-MSN and cause cell-death, which leads to comparable results.

The question is relevant to the field and the methodology used to address it is novel and appropriate. The findings are interesting and some how opposite from previous published results, which is interesting and important. However, it is critical to validate the cell-specificity of the ablation/opto inhibition with regards to cholinergic interneurons in order to strengthen the interpretation of the results and the conclusion of the study.

Main comments:

1. The title of the study refers to "ventrolateral striatum". However the ablation mainly affects the ventral striatum (nucleus accumbens) and the dorsomedial striatum, not the dorsolateral. This should be revised in the title and throughout the study.
2. The evidence that cholinergic interneurons are not targeted in D2-DTA mice is critical for the interpretation of the results. If these interneurons, which are well-known to express D2Rs, were affected by the ablation, it could explain the difference in phenotype observed here (decreased breakpoint) and that contradicts previous literature. Currently, this evidence is only provided for the D2-DTA mice. It is important to do the same in the D2-ArchT mice. Please report expression level of ArchT-GFP in the cholinergic interneurons and in MSN in the D2-ArchT mice. This is a new mouse line and expression pattern could be different from D2-DTA mice.
3. On this same issue, Fig. 1G shows in situ for ChAT and *Drd2* (not D2R, please fix) mRNA but it seems like they do not correspond to same section and thus colocalization can be not quantified. It is important to add the quantification of the colocalization experiments between ChAT and DTA mRNA as it will strengthen the evidence that the interneurons are spare in this manipulation.
4. Also, the current quantification corresponds to density of ChAT-positive neurons. Please express the density of neurons as cell/area of tissue, not per section as it can vary from section to section.
5. The selection of controls for the in vivo electrophysiology experiment is questionable and not ideal. WT mice are used here and there are differences compared to D2-DTA mice after 7 days of DOX-OFF. However, the more appropriate controls are D2-DTA mice while in DOX treatment. How stereotypic is the proportion of responses types obtained?
6. On those same experiments, are there differences in the baseline firing of VP neurons after ablation

of D2-MSN? It could be expected that changes in baseline would develop.

7. Also, why does the frequency of eex-inh pattern goes up with the treatment. If the inhibition phase 2 corresponds to the VSL-VP connections, those should have been down, unless there is compensation from inhibition arising from other neurons. And then this increase inhibition from VSL could also account for the behavioral changes observed or lack of.

8. Optogenetic inhibition/ablation approach. These are very nice and important set of experiments as they provide an independent validation of the main findings. It is mentioned that GFP-positive neurons were found in the midbrain. Actually the statement is confusing and it reads "a few GFP-positive cells at dopamine neurons,..." Does this mean co-label with dopamine neuron markers? Please state clearly and show data in supplementary figure. Also, low fluorescence levels "suggest" rather than "indicate" that optogenetic inhibition will not affect dopamine levels in the accumbens. If the authors wish to make this statement stronger and want to show this, then data will need to be added (e.g> electrophys recording from dopamine neurons or dopamine measurements in the region of the fiber implantation, etc).

Other minor comments:

9. The authors used the term bigenic and monogenic. Are they referring to homozygote and heterozygote? Is there a good reason why not to use those terms?

10. Please add a reference for the statement of long-term ArchT activation leading to cell death.

11. I suggest moving the data presented in Fig S4 to the main figures. The quantification and correlation of the degree of Drd2 mRNA loss and the breakpoint reduction is important and contributes significantly to the understanding of the circuit that control motivated behavior.

12. With regards to the striatal region specificity of the cell-ablation observed in D2-DTA mice upon DOX-OFF treatment, is it possible that it levels of Drd2 mRNA expression in the different regions have something to do with the higher sensitivity of Drd2 expression in the ventral and dorsomedial region of the striatum. Can the authors detect any correlation with the pattern of expression for Drd2 mRNA WT mice?

Reviewer #2 (Remarks to the Author):

Summary:

The authors perform cell type specific ablation of D2R expressing neurons in the ventral striatum, and show that mice have lower motivation on an operant task. They followed up these results with elegant studies using optogenetics and a novel "optogenetic ablation". I found the manuscript interesting and the data high quality. However, I have a few comments that should be addressed to fully support their conclusions.

Major comments:

1. I did not find the description of their behavioral effects as "apathy" helpful. Apathy is a conscious state in humans that seems very difficult to model in mice. More commonly, the behaviors they tested are described as tests of "motivation", and defined operationally. Is there a reason why the author's don't see their experiments as testing motivation?

2. The electrophysiological experiments in Figure 3 conclude that remaining living D2R-expressing

neurons are hypofunctioning in DOX-off day 7 mice. However, they are recording VP neurons that receive input from hundreds of MSNs. Therefore, it seems equally likely that the DTA-exposed but living D2R-expressing neurons are in fact normal, but there are just fewer of them due to the ablation, hence the weaker inhibitory responses in the VP. I don't see how the authors can dissociate these points via in vivo recordings. To properly evaluate these possibilities the authors should use slice recordings from D2R expressing MSNs. However, I also don't think it's critical to their conclusions that the remaining neurons are hypo-functioning, so they could remain agnostic on this point and report both possibilities.

3. The classification of Phases I, II, and III in the 5-CSRTT seems arbitrary, with each phase containing a different number of days in a way that appears to allow Phase II to capture the days when the data appeared significant. Was a rationale approach used to define these phases that I'm just missing? If not, it would be more appropriate to report which specific days were significant, controlling for multiple comparisons with a Benjamini-Hochberg False Discovery Rate that will protect against false negatives due to the high number of comparisons.

4. The optogenetic ablation experiment is extremely interesting, and a potentially novel and useful application of optogenetics. However, I am not convinced they achieved ablation from the data they report. They show loss of GFP, microglial activation and reductions in D2R mRNA, none of which is directly linked to cell death. I'd be more convinced by NeuN staining showing fewer living neuronal nuclei, or another stain that specifically evaluates cell death.

Minor comments:

1. In several places the authors include discussion and interpretation within the results, and at times I felt it was too much. Most notably, when discussing emotional regulation and anhedonia (Lines 204-206). While this interpretation is interesting, it should be moved to the discussion due to its speculative nature.

2. Certain behavioral details were missing. In particular, the time of day when experiments were run was not given, and is important given the food-based operant responding that was used as an assay of motivation. In addition, it is unclear whether the mice undergoing the various behavioral tasks in Figure 4 are the same mice or different mice.

3. In Figure 1 the authors show data ruling out non-specific toxicity on ChAT neurons and dopaminergic neurons, but put the data on D1R-expressing neurons into supplemental figure 2. I would put this in the main figure, as it argues against a non-specific toxicity that is difficult to evaluate from ChAT and dopamine neurons.

4. Figure 2 shows methods that could be placed in a supplemental figure.

Reviewer #3 (Remarks to the Author):

In this paper, the authors characterized the functional role of ventrolateral striatal D2 receptor-expressing neurons in goal-directed behavior, and argue that ablation of this specific population of neurons results in an increase in apathy. They generated a new D2-tTA line, which they crossed with a tetO-DTA line, and used this line to show that a progressive destruction of D2 neurons (spreading from ventrolateral striatum to more dorsal regions of the striatum) differentially affects motivated behavior depending on the amount of destruction. Early timepoints following DOX removal (ventrolateral striatum damage) result in an apathetic-like phenotype in goal directed behavior (3 choice serial reaction time task and progressive ratio task). Later timepoints following DOX removal

(ventral and some dorsal striatal damage) result in an inability to withhold responding (premature response in the 3 choice serial reaction time task) and an increase in locomotor activity in the open field task. Last, optogenetic inhibition and ablation of ventrolateral striatal D2 neurons leads to deficits in the progressive ratio task. The study is interesting, well controlled, and well written. I have some specific suggestions for improvement:

The authors have generated a new mouse line with tTA targeted to D2 neurons. They have characterized this line by crossing it with a tetO-ChR2 line, but the supplementary figures depicting this characterization are currently somewhat unclear. Could the authors provide quantification of both specificity and penetrance in D2 neurons? It is unclear which is depicted.

The central hypothesis is that loss of D2 neurons in ventrolateral striatum leads to apathy (e.g. Fig 4b, Fig 5b) while the spread of this loss to more dorsal regions of the striatum leads to deficits in inhibiting movements (e.g. Fig 2b, Fig S3). The authors have demonstrated that optogenetic inhibition of D2 neurons in the VLS leads to a reduction in breakpoint in the progressive ratio task. It would be helpful to bolster this claim by optogenetically inhibiting D2 neurons in more dorsal regions to demonstrate (for example) increased premature responding.

The authors have a bigenic D2-tTA::tetO-ChR2 mouse in their lab. What are the effects of optogenetic activation of D2 neurons in ventrolateral and more dorsal striatum?

There are D2-expressing neurons in the cortex, in particular layer 5 neurons in the medial prefrontal cortex. Dopamine in this region is hypothesized to play a role in apathy. It would be straightforward for the authors to provide an anatomical characterization of DTA mRNA and the loss (or not) of D2 neurons in this region as a time series following DOX off as in Figure 1.

Please plot individual animals (potentially as dots) on all figures with bar graphs (e.g. Figure 2b, 4f, 4g, etc).

Reviewer #4 (Remarks to the Author):

This paper examines the effect of a conditional ablation of D2-MSNs in the ventral striatum on behaviour. They find significant loss of D2 mRNA in the ventral striatum after 10 days of removal of the tet suppression of DTA. They observe a lasting increase in locomotor activity, decrease in effort-based instrumental action, increase in impulsivity and compulsivity and a transient effect on cognition in a 3CSRTT. I think they have developed a very interesting model and their data supports some experimental findings that are previously published. For example, decreased D2 receptor predict increased trait impulsivity (Dalley et al., 2007 Science), Activation of D2 receptor expression induces bradykinesia (Kravitz et al., 2010 Nature), where as deletion of D2 in iMSNs induces hyperlocomotor activity, deficits in spontaneous movement and motor skill performance (Lemos et al., 2016 Neuron). Further viral knockdown of D2Rs increases reward threshold on intra cranial self stimulation (Johnson et al 2010, Nat Neurosci.). Here, they characterize a novel method of conditionally knocking down MSNs expressing D2 receptors in the ventral striatum and attempt bring together some of these ideas in a cohesive hypothesis. However, some of their interpretations may have alternate explanations.

First, the authors propose that ablation of iMSN D2 receptors in the Ventral striatum induces apathy or a state of amotivation. The evidence they sue to support this is that they see a reduction in goal directed behaviour though a decrease in trials and increase in omissions observed 6-7 days after removal of the tet suppression. Similarly the impairment in effort-based instrumental responding (PR) occurred within 3-4 days after removal of tet suppression. At this timepoitn they see expression of

DTA mRNA, but no alterations in Drd2 mRNA expression until later (after day 10 of tet off). These timepoints of the behavioral alterations fit within the timeframe of an immunological response (activated microglia), but not necessarily within the timeframe of Drd2 loss. Therefore, I would interpret their alterations in motivated/goal-directed behaviour would be likely due to an inflammatory response rather than loss of D2 receptors.

The behavioural data that fits best with the timeframe of loss of D2 receptors is the increased impulsivity and compulsivity on the 3CSRRT (in the supplemental) along with the alterations in locomotor activity. This also supports previous reports of hyperlocomotor activity (Lemos et al., 2016) and increased impulsivity (Dalley et al., 2007) with loss of D2Rs. However, this does not support their hypothesis that decreased D2Rs result in apathy.

If I understand this experiment correctly, to obtain D2-ArchT biogenic mice, they presumably crossed Drd2-tTA mice with TetO-ArchT-EGFP mice (Additional information on this should be in the methods rather than just a reference to the orexin/hypocretin paper in the results). However, I am unclear how this strategy targets ArchT-EGFP only to the D2R expressed in MSNs and not to all DR2 expressing cells. They indicate that they observed little ArchT-EGFP fluorescence in the VTA dopamine neurons. However, this does not exclude the D2 receptors expressed on glutamatergic inputs or cholinergic inputs to MSNs in the ventral striatum. Presumably inhibition (or light-induced ablation) of these D2-expressing inputs would alter goal-directed behaviour. Can they demonstrate (or further explain) how this targets only MSN D2 receptors? Furthermore, could their 3h photostimulation to ablate the D2 expressing cells result in changes in neuroinflammation?

Minor:

Fig S2c - numbers in the table are way too small to see

Data in Fig 3 should be included in the Fig 4, - to save room, move data in 4F,G,H to the supplemental.

Line 304 - they are not really looking at reward value, rather reward preference

Fig S3c - for sensitization they need to test if locomotor activity on day 5 is greater than that on Day in both groups.

Fig. S4d. They should label the units for the preference score on the y axis.

We would like to thank the reviewers for their careful reading of our manuscript and their thoughtful comments. Their suggestions are greatly appreciated and nearly all of them have been incorporated into the revised manuscript. Please find our point-by-point responses to the reviewers below. The reviewers' comments are numbered, underlined, and in italics. Our revised sentences are in the bold face.

Reviewer #1

Comment #1. The title of the study refers to "ventrolateral striatum". However the ablation mainly affects the ventral striatum (nucleus accumbens) and the dorsomedial striatum, not the dorsolateral. This should be revised in the title and throughout the study.

It is true that DTA-mediated cell dysfunction/ablation area covered the ventral striatum and the dorsomedial striatum eventually. However, our experiments were designed to elucidate the effects of D2-MSNs ablation in the "ventrolateral striatum". We believe the DOX-off and re-start regimen enabled us to (temporarily) confine the cell dysfunction area within the VLS (Fig. 5).

We would like to use the term "ventrolateral" instead of "nucleus accumbens" because DTA-mediated cell dysfunction area was not limited to the rostral part of the striatum (probably including the lateral part of the accumbens core and lateral shell) but located from the rostral to the caudal part of the striatum (Fig. 2A).

According to above two reasons, we believe (and hope the reviewer agrees) that the term "ventrolateral striatum" more accurately depicts the region related to the main topic of this study.

Comment #2. The evidence that cholinergic interneurons are not targeted in D2-DTA mice is critical for the interpretation of the results. If these interneurons, which are well-known to express D2Rs, were affected by the ablation, it could explain the difference in phenotype observed here (decreased breakpoint) and that contradicts previous literature. Currently, this evidence is only provided for the D2-DTA mice. It is important to do the same in the D2-ArchT mice. Please

report expression level of ArchT-GFP in the cholinergic interneurons and in MSN in the D2-ArchT mice. This is a new mouse line and expression pattern could be different from D2-DTA mice.

We agree with reviewer's comment. The reviewer is concerned whether *Drd2*-mRNA positive cholinergic interneurons expressed DTA in D2-DTA mice. To solve this concern, we added the data with double fluorescent *in situ* hybridization for *ChAT* (the marker of cholinergic neurons) and *DTA* mRNA. *ChAT*-positive cells were never labeled with *DTA* (none out of 100 ChAT positive cells, from 2 brains), indicating that cholinergic interneurons were not targeted in this DTA experiments. We added this new data (please see below) in our revised manuscript (Figure 1I).

We revised the result section as follows:

Before:

The numbers of *Drd2* mRNA-positive striatal cholinergic interneurons and dopaminergic neurons did not change after DOX removal (Figures 1G). Indeed, *DTA* mRNA was not detectable in dopaminergic neurons (data not shown).

After (page 4, line 29):

The numbers of *Drd2* mRNA-positive striatal cholinergic interneurons and dopaminergic neurons did not change after DOX removal (**Figures 1G and 1J**). Indeed, **after DOX removal, *DTA* mRNA was not detectable in cholinergic interneurons (Figure 1I)** or in dopaminergic neurons (data not shown).

Regarding the expression of ArchT-EGFP in cholinergic interneuron in D2-ArchT mice, we conducted double immunohistochemistry with GFP and CHT1 (choline transporter1, which is also the marker of cholinergic neurons) to examine the

penetrance of *Drd2* promoter-mediated gene induction. None of CHT1 positive cholinergic interneurons expressed GFP (zero out of 70 cells), supporting the evidence that *tTA* mRNA did not express in *Drd2*-tTA mice.

We examined ArchT-EGFP expression in MSNs in the striatum and found that D2-MSNs specific EGFP expression. We provided the information regarding ArchT-EGFP expression of the cholinergic interneuron and the MSNs in Supplemental Figure S5 and Table S1.

We added the following sentences to describe ArchT-EGFP specific expression in the D2-MSNs in the result section.

Page 8 line 33:

***Drd2*-positive cholinergic interneurons were not labeled with GFP (Figure S5C). The pyramidal neurons are known to express *Drd2* mRNA, and their axon terminals project to the striatum; however, neurons in the medial prefrontal cortex and IC were not labeled with GFP (Figure S5E). Together with these data, ArchT expression within the striatum was specific to the D2-MSNs (Table S1).**

Comment #3. On this same issue, Fig. 1G shows in situ for ChAT and *Drd2* (not *D2R*, please fix) mRNA but it seems like they do not correspond to same section and thus colocalization can be not quantified. It is important to add the quantification of the colocalization experiments between ChAT and *DTA* mRNA as it will strengthen the evidence that the interneurons are spare in this manipulation.

We displayed the confocal images in the previous Fig. 1G (now Figures 1H and 1I). To clarify this, we added the method information in the Figures 1H and 1I legends in our revised manuscript (page 28, line 35, and page 29, line 3).

We conducted double fluorescent in situ hybridization for *ChAT* and *DTA* mRNA at DOX off day 10 and found that none of *ChAT* positive cells expressed *DTA* mRNA (please see comment #1). We described this histological data in the result section (page 4, line 25) and added the quantification in Figure 1I legend (page 29, line 4).

Comment #4. *Also, the current quantification corresponds to density of ChAT-positive neurons. Please express the density of neurons as cell/area of tissue, not per section as it can vary from section to section.*

We described the density of ChAT-positive neurons as cell/area in our revised manuscript (**Figure 1G**) as follows.

Comment #5. *The selection of controls for the in vivo electrophysiology experiment is questionable and not ideal. WT mice are used here and there are differences compared to D2-DTA mice after 7 days of DOX-OFF. However, the more appropriate controls are D2-DTA mice while in DOX treatment. How stereotopic is the proportion of responses types obtained?*

According to reviewer's advice, we conducted electrophysiological analysis by using D2-DTA with DOX on regimen and obtained comparable data with previous experiment using WT mice. We replaced the data in our revised manuscript (**Figure 3 and Table 1**).

The recording sites were reconstructed in all cases by the probe track, which was visualized with DiI. The depth of the recorded neurons was evaluated with the distance from the dura. Responsive neurons were found in the same region, and each response pattern was randomly obtained in the VP of ON, OFF7 and OFF20. We included the random distribution of responded neurons in the method section as follows.

Page 15, line 2:

Each response pattern was randomly obtained in the ventral pallidum in DOX on and off regimens.

Comment #6. *On those same experiments, are there differences in the baseline firing of VP neurons after ablation of D2-MSN? It could be expected that changes in baseline would develop.*

The baseline firings (count/s) of VP was comparable between groups (DOX on control: 47.3 ± 0.3 , DOX off days 7: 52.0 ± 0.3 , DOX off days 20: 50.9 ± 0.3) in this study.

Although the precise mechanism is unknown, this result is consistent with the previous report with immunotoxin-mediated D2-MSNs ablation study (Sano et al., 2013, ref 35).

We now include this information in the method section.

Page 15, line 17:

In both DOX-on and –off periods, baseline firing of VP were comparable as previous immunotoxin-mediated cell ablation study reported³⁵.

Comment #7. *Also, why does the frequency of eex-inh pattern goes up with the treatment. If the inhibition phase 2 corresponds to the VLS-VP connections, those should have been down, unless there is compensation from inhibition arising from other neurons. And then this increase inhibition from VLS could also account for the behavioral changes observed or lack of.*

As the reviewer pointed out, the ratio of eex-inh pattern at OFF20 increased compared with that at OFF7 although the degree of MSN dysfunction was comparable. One plausible explanation is that D1- MSNs in the VLS-VP pathway compensate the inhibition at OFF20. However, this explanation does not account for the same response patterns including late excitation (eex-inh-lex and eex-lex), which corresponds to VLS-VP-STN-VP pathway.

It is difficult to solve this specific concern (*why does the frequency of eex-inh pattern goes up with the treatment?*), but we believe that we convince the readers that DOX-off treatment (both OFF7 and OFF20) resulted in the decreased responses containing inhibition phase according to this population histogram.

Regarding the behavioral consequence at OFF20, the effects from extended area (VMS and DMS) should be added to that from the VLS. Thus it is difficult to address the later concern (*this increase inhibition from VLS could also account for the behavioral changes observed or lack of*). Please consider these limitations.

Comment #8. Optogenetic inhibition/ablation approach. These are very nice and important set of experiments as they provide an independent validation of the main findings. It is mentioned that GFP-positive neurons were found in the midbrain. Actually the statement is confusing and it reads "a few GFP-positive cells at dopamine neurons...." Does this mean co-label with dopamine neuron markers? Please state clearly and show data in supplementary figure. Also, low fluorescence levels "suggest" rather than "indicate" that optogenetic inhibition will not affect dopamine levels in the accumbens. If the authors wish to make this statement stronger and want to show this, then data will need to be added (e.g> electrophys recording from dopamine neurons or dopamine measurements in the region of the fiber implantation, etc).

To clarify what GFP positive cells were in the midbrain, we conducted a double immunohistochemistry with GFP and TH (the marker of dopamine neurons) (2 D2-ArchT mice, 4 sections). We found that GFP-immunopositive midbrain neurons were TH-positive dopamine neurons in the VTA; 90% of GFP-positive cells ($n= 7.0 \pm 2.3$) were labeled with TH and 7% of TH-positive cells ($n=76.8 \pm 8.6$) were labeled with GFP. We added these in Supplemental Figure S5C.

Regarding the assumption of ArchT functional expression in DA neurons, we rephrased the term per the reviewer's comment.

Before:

Immunohistochemistry detected a few GFP-positive cells at dopamine neurons, however, the level of GFP was too low to observe direct fluorescence (data not shown), indicating that optogenetic inhibition should not work in dopamine neurons due to the low level of ArchT expression.

After (page 8, line 30):

Immunohistochemistry detected a few GFP-positive dopamine neurons (**Supplementary Figure S5C**), however, the level of GFP was too low to observe direct fluorescence (data not shown), **suggesting** that optogenetic inhibition should not work in dopamine neurons due to the low level of ArchT expression.

Comment #9. *The authors used the term bigenic and monogenic. Are they referring to homozygote and heterozygote? Is there a good reason why not to use those terms?*

The Tet system is a bipartite system; the system requires two distinct lines, tTA and tetO lines. Therefore, the researchers use bigenic (double transgenic) and monogenic (single transgenic) instead of hetero- and homozygote.

Comment #10. *Please add a reference for the statement of long-term ArchT activation leading to cell death.*

To our knowledge, there is no previous report describing the opto-ablation.

Comment #11. *I suggest moving the data presented in Fig S4 to the main figures. The quantification and correlation of the degree of Drd2 mRNA loss and the breakpoint reduction is important and contributes significantly to the understanding of the circuit that control motivated behavior.*

We moved the data to **Figure 5F** as the reviewer suggested.

Comment #12. *With regards to the striatal region specificity of the cell-ablation observed in D2-DTA mice upon DOX-OFF treatment, is it possible that the levels of Drd2 mRNA expression in the different regions have something to do with the higher sensitivity of Drd2 expression in the ventral and dorsomedial region of the striatum. Can the authors detect any correlation with the pattern of expression for Drd2 mRNA WT mice?*

We examined the differences of the *Drd2* mRNA expression level within the striatum by *in situ hybridization*. ISH is not a qualitative method, however, it can qualitatively address mRNA level in the single cell level. Especially, at the beginning of the color development, cells with higher mRNA were labeled weakly and those with lower mRNA was not. As shown in below pictures (a: low magnification, *Drd2* ISH, 30 min development, without nuclear fast red stain, b: dorsolateral, c: ventrolateral), there was no regional difference of *Drd2* mRNA level. We think that the regional difference of *Drd2* mRNA level is unlikely the cause of the preferential targeting of VLS in our system.

Reviewer #2

Comment #1. *I did not find the description of their behavioral effects as "apathy" helpful. Apathy is a conscious state in humans that seems very difficult to model in mice. More commonly, the behaviors they tested are described as tests of "motivation", and defined operationally. Is there a reason why the author's don't see their experiments as testing motivation?*

We realize that modeling apathy in animals is controversial, however, we continue to feel that our use of the term is justified. First, our finding that striatal neurons mediate

decreased motivation in mice is true. The etiology (striatal lesion) and resultant decreased motivation provide construct and face validity for a model of human apathy. Second, apathy is a pervasive clinical phenomenon that deserves more attention at the translational and pre-clinical levels. Our hope is that our findings will help generate interest in understanding how animal studies of motivation can shed light on the human phenomenon. We would be very happy if reviewer #2 can now accept our link between apathy and decreased motivation in mice.

Comment #2. *The electrophysiological experiments in Figure 3 conclude that remaining living D2R-expressing neurons are hypofunctioning in DOX-off day 7 mice. However, they are recording VP neurons that receive input from hundreds of MSNs. Therefore, it seems equally likely that the DTA-exposed but living D2R-expressing neurons are in fact normal, but there are just fewer of them due to the ablation, hence the weaker inhibitory responses in the VP. I don't see how the authors can dissociate these points via in vivo recordings. To properly evaluate these possibilities the authors should use slice recordings from D2R-expressing MSNs. However, I also don't think it's critical to their conclusions that the remaining neurons are hypo-functioning, so they could remain agnostic on this point and report both possibilities.*

The reviewer raised possibility that the net effect observed in the electrophysiology in early time points of DOX off regimen (e.g. DOX off days 7) was mediated via cell ablation (dead cells). However, we did not detect any dead cells in the VLS at DOX off days 7 (Figure 1F), indicating that the net effect was unlikely to be mediated via dead cells. We think that it is reasonable to interpret that altered *in vivo* electrophysiology results was mediated via hypofunctioning viable DTA-exposed cells.

To clarify that D2-MSNs did not die at this time point, we added the phrase as follows.

Page 5, line 13:

Our histological analysis revealed that cell death and apparent loss of *Drd2* mRNA occurred after DOX-off day 10. However, **prior to cell death**, *DTA* mRNA was expressed at earlier times (DOX off for 3–7 days) (Figures 1B and 1F).

Comment #3. *The classification of Phases I, II, and III in the 5-CSRTT seems arbitrary, with each phase containing a different number of days in a way that appears to allow Phase II to capture the days when the data appeared significant. Was a rationale approach used to define these phases that I'm just missing? If not, it would be more appropriate to report which specific days were significant, controlling for multiple comparisons with a Benjamini-Hochberg False Discovery Rate that will protect against false negatives due to the high number of comparisons.*

As the reviewer #2 pointed out, we did not clearly explain the rationale of our classification. We classified the periods based on the results of histological analysis: the phase before loss-of-function manipulation is classified as Phase I, the phase from the timing of *DTA* mRNA appearance in the VLS (DOX off days 3) to the timing of cell death appearance (DOX off days 10) as Phase II, and the phase after the cell death expansion to the whole VS (DOX off days 14~) as Phase III.

We re-analyzed our data and realized that the increase in %omission was not significant (phase: $F_{2, 15} = 2.943$, $P = 0.056$, phase \times group interaction: $F_{2, 20} = 1.954$, $P = 0.061$) in D2-DTA animals (Figure 4B). Accordingly we revised statistical data and related sentences as follows.

Before:

Following DOX-off conditions for 5 days, D2-DTA bigenic mice displayed a decreased total number of trials in 60 min of testing (phase from days 5 to 10: $F_{2, 15} = 9.624$, $P = 0.002$; phase \times group interaction: $F_{2, 20} = 4.811$, $P = 0.024$, Figure 4B) and an increased ratio of omission responses to total trials (%omission) (phase: $F_{2, 15} = 10.543$, $P = 0.001$; phase \times group interaction: $F_{2, 20} = 6.954$, $P = 0.007$, Figure 4C) compared to monogenic controls (total trial: $t_{10} = 2.445$, $P = 0.04$, Figure 4B; %omission: $t_{10} = 2.758$, $P = 0.03$, Figure 4C). D2-DTA bigenic mice at DOX off day 10 showed normal locomotor activity (Figure 2A), supporting the idea that the decreased number of total trials achieved was caused by impairment of instrumental motivation (Robbins, 2002). Increased %omission was likely due to the reduction of sustained motivation, rather than reduction of sustained attention, since other parameters representing cognitive

activities were intact (%accuracy, phase: $F_{2, 15} = 2.749$, $P = 0.094$, *NS*, phase \times group interaction: $F_{2, 20} = 0.294$, $P = 0.752$, *NS*, Figure 4D; correct response latency, phase: $F_{2, 15} = 1.80$, $P = 0.09$, *NS*; phase \times group interaction, $F_{2, 20} = 0.681$, $P = 0.520$, *NS*, Figure 4E) in this study (Robbins, 2002).

After (page 6, line 25):

Following DOX-off conditions for 3 days, D2-DTA bigenic mice displayed a decreased total number of trials in 60 min of testing (phase from days 3 to 10: $F_{2, 15} = 9.102$, $P = 0.004$; phase \times group interaction: $F_{2, 20} = 4.832$, $P = 0.022$, Figure 4B) compared to monogenic controls (total trial: $t_{10} = 2.422$, $P = 0.045$, Figure 4B). D2-DTA bigenic mice at DOX off day 10 showed normal locomotor activity (Figure 2A), supporting the idea that the decreased number of total trials achieved was caused by impairment of instrumental motivation (Robbins, 2002). **There was a trend of increased %omission (phase: $F_{2, 15} = 2.943$, $P = 0.056$, *NS*, phase \times group interaction: $F_{2, 20} = 1.954$, $P = 0.061$, *NS*, Figure 4C), which** was likely due to the reduction of sustained motivation, rather than reduction of sustained attention, since other parameters representing cognitive activities were intact (%accuracy, phase: $F_{2, 15} = 2.692$, $P = 0.095$, *NS*, phase \times group interaction: $F_{2, 20} = 0.301$, $P = 0.731$, *NS*, Figure 4D; correct response latency, phase: $F_{2, 15} = 1.992$, $P = 0.089$, *NS*; phase \times group interaction, $F_{2, 20} = 0.699$, $P = 0.493$, *NS*, Figure 4E) in this study.

We added the following explanation of time classification in the method section:

Page 18, line 11:

The behavioral data were analyzed in three phases: the phase before loss-of-function manipulation is classified as Phase I; the phase from appearance of DTA mRNA in the VLS (DOX off days 3) to the appearance of cell death (DOX off days 10) as is classified as Phase II; and the phase after the cell death expansion to the whole VS (DOX off days 14~) is classified as Phase III.

We also re-analyzed the behavioral data of PR experiment in accordance with the new classification (Figure 5) and obtained similar results as previous. We revised the result section as follows.

Page 7, line 30:

D2-DTA bigenic mice started to display a behavioral reduction after DOX was off for 3 days and this reduction further deteriorated day-by-day according to decreased break points (phase \times group interaction: $F_{2,40} = 5.782$, $P = 0.021$, Figure 5B) and prolonged time spent to complete the PR task (phase \times group interaction: $F_{2,40} = 7.344$, $P = 0.003$, Figure 5C). These observations were not evident in controls (break point: $t_{20} = 13.211$, $P = 0.005$, time spent to complete the PR: $t_{20} = 15.899$, $P = 0.002$, Figures 5B and 5C). After the DOX restart, the behavioral reduction remained (break point, *post hoc* analysis between groups at phase III: $t_{20} = 17.377$, $P = 0.004$, Figure 5B; time spent to complete the PR, *post hoc* analysis between groups at phase III: $t_{20} = 21.093$, $P = 0.002$, Figure 5C). Associative learning and appetite were unaffected (Figures 5D and 5E) as seen in the 3-CSRTT (Figures 4D and 4E), suggesting that cognitive and emotional dimensions were spared.

Comment #4. *The optogenetic ablation experiment is extremely interesting, and a potentially novel and useful application of optogenetics. However, I am not convinced they achieved ablation from the data they report. They show loss of GFP, microglial activation and reductions in D2R mRNA, none of which is directly linked to cell death. I'd be more convinced by NeuN staining showing fewer living neuronal nuclei, or another stain that specifically evaluates cell death.*

We conducted a new experiment and confirmed the 3-hr illumination of the ArchT expressing D2-MSNs induced cell death by observing the expression of single strand DNA (ssDNA) and the decreased number of NeuN positive cells (Control: 62.8 ± 3.6 , Opt-ablation: 29.3 ± 3.8 , mean \pm S.E.M., cells/area, 4 area) beneath the tip of the optical fiber. We believe that added data solve the reviewer's concern. We added this data in **Supplemental Figure S6** and revised the sentence as follows.

Figure S6, Related to Figure 6

Optogenetic ablation in D2-ArchT mice.

Legend: The 3-hr yellow light (upper panels) illumination induced cell death, which was supported with the expression of single strand DNA (ssDNA) and the decreased number of NeuN positive cells (Control: 62.8 ± 3.6 , Opt-ablation: 29.3 ± 3.8 , mean \pm S.E.M., cells/white dash square, 4 squares). White dash square = $400 \mu\text{m} \times 400 \mu\text{m}$. White arrows indicate the tip of optic fibers. Scale= $20 \mu\text{m}$

Page 9, line 19:

Such illumination resulted in the reduction of GFP immunoreactivity below the tip of the fiber, **the appearance of ssDNA, the decreased number of NeuN positive cells,** the apparent loss of *Drd2* mRNA positive medium-size cells (while sparing *Drd1* mRNA positive cells), and the activation of microglial cells, suggesting a D2-MSN-specific ablation (Figure 6K and **Figure S6**).

Comment #5. *In several places the authors include discussion and interpretation within the results, and at times I felt it was too much. Most notably, when discussing emotional regulation and anhedonia (Lines 204-206). While this interpretation is interesting, it should be moved to the discussion due to its speculative nature.*

Considering this comment and the comment from another reviewer (Reviewer #4, comment #5), we moved figures presenting emotional regulation and food preference/consumption (previous Figure 4F-H) to Supplemental Figure S3C-E.

According to the change, we removed the corresponding paragraph from the result, but we keep the data with food preference/intake in the result section because this data should be provided in food-incentive instrumental tasks. We have made a change as follows.

Before:

.....Given these alterations in behavior, loss-of-function of ventrolateral D2-MSNs induces quantitative reductions in goal-directed behavior, which can be interpreted as decreased instrumental motivation.

Mice experiencing the DOX-off day 7 regimen displayed a comparable degree of anxiety-like behavior in the elevated plus maze test (Total distance: $t_{16} = 1.201$, $P = 0.25$, NS; Open arm spent time: $t_{16} = 0.808$, $P = 0.43$, NS, **Figure 4F**), a comparable degree of despair-related behavior in the forced swim test (Immobility: $t_{16} = 0.790$, $P = 0.44$, NS; Climbing: $t_{16} = 1.542$, $P = 0.14$, NS, **Figure 4G**), and a comparable degree of anhedonia-like behavior (Palatable food preference, $F_{4,40} = 0.759$, $P = 0.559$; Palatable food consumption, $F_{4,40} = 0.357$, $P = 0.837$, NS, **Figure 4H**), suggesting that emotional dysregulation was not involved in decreased motivation at this time point.

After (page 7 line 3):

.....Given these alterations in behavior, loss-of-function of ventrolateral D2-MSNs induces quantitative reductions in goal-directed behavior, which can be interpreted as decreased instrumental motivation. **Mice experiencing the DOX-off day 7 regimen**

displayed a comparable food preference/intake (Figure S3), strengthening the selective effect of loss-of-function of D2-MSN on food-incentive instrumental tasks.

Before:

Importantly, reward preference (Figures 4E, 5E, and 6I), associative learning (Figures 4D, 5D, and 6H), and spontaneous behavior (Figures 2B and 4F) were not altered by D2-MSN dysfunction, suggesting that D2-MSN dysfunction specifically impairs goal-directed behavior.

After (page 10, line 7):

Importantly, reward preference (Figures 4E, 5E, 6I, **and Figure S3A**), **emotional regulation (anxiety-like behavior [Figure S3B], despair-like behavior [Figure S3C])**, associative learning (Figures 4D, 5D, and 6H), and spontaneous behavior (Figures 2B and 4F) were not altered by D2-MSN dysfunction, suggesting that D2-MSN dysfunction specifically impairs goal-directed behavior.

Comment #6. Certain behavioral details were missing. In particular, the time of day when experiments were run was not given, and is important given the food-based operant responding that was used as an assay of motivation. In addition, it is unclear whether the mice undergoing the various behavioral tasks in Figure 4 are the same mice or different mice.

All the behavioral experiments were conducted during the light phase (12:12-h light/dark cycle; lights on at 8 am). We added this sentence in the method section of our revised manuscript.

Page 13, line 23:

All mice were maintained with 12:12-h light/dark cycle (lights on at 8 am) and the behavioral experiments were conducted during the light phase.

We included 3-CSRTT, EPM, FST, Food preference test, Food consumption test in previous Figure 4. Among these, we used three cohorts; 1) 3-CSRTT, 2) EPM and FST, 3) Food preference and Food consumption.

As the reviewer #4 recommended, we now move data of EPM, FST, Food preference test, and Food consumption test to supplementary Figure S3. We described the information if the same mice were used in the different tests (supplementary, page 14).

In relation to this comment, we realized that we did not describe the method for food preference/consumption tests. We now include it in supplementary page 14.

Comment #7. *In Figure 1 the authors show data ruling out non-specific toxicity on ChAT neurons and dopaminergic neurons, but put the data on D1R-expressing neurons into supplemental figure 2. I would put this in the main figure, as it argues against a non-specific toxicity that is difficult to evaluate from ChaT and dopamine neurons.*

We added the data showing that D1-MSNs were spared in D2-DTA mice in **Figure 1K** and added the description as follows:

Page 4, line 29:

The number of dopamine receptor type 1-expressing medium spiny neurons (D1-MSNs) in the VLS did not change after DOX removal (Figure 1K).

Comment #8. *Figure 2 shows methods that could be placed in a supplemental figure.*

We would like to keep this figure in the main figure. It is very important information that the ablation is not limited to the rostral part, which is related to the reply to the comment #1 from the reviewer #1.

Reviewer #3

Comment #1. *The authors have generated a new mouse line with tTA targeted to D2 neurons. They have characterized this line by crossing it with a tetO-ChR2 line, but the supplementary figures depicting this characterization are currently somewhat unclear. Could the authors provide quantification of both specificity and penetrance in D2 neurons? It is unclear which is depicted.*

We provide the supplementary table summarizing the specificity of tTA expression and the penetrance of tTA-mediated gene induction in both D2-DTA and D2-ArchT mice. We believe that this table helps readers to understand *Drd2*-tTA mediated targeting.

Table S1

			tTA expression	DTA induction	ArchT-EGFP induction
Striatum	Drd2 -positive cells	D2-MSNs	+	+ (Fig 1B)	+ (Fig S5B)
		Cholinergic interneurons	- (Fig 1H)	- (Fig 1I)	- (Fig S5D)
	Drd2 -negative cells	D1-MSNs	-	- (Fig 1K)	- (Fig S5B)
Outside striatum	Drd2 -positive cells	DA neurons	+	- (Fig 1J)	+/- (Fig S5C)
		IC neurons	-	- (Fig S1F)	- (Fig S5E)
		mPFC neurons	-	- (Fig S1F)	- (Fig S5E)
		Mossy cells	-	-	-

Note: +: detected, -: not detected, +/-: detected but unlikely functional, parenthesis shows related data.

Comment #2. The central hypothesis is that loss of D2 neurons in ventrolateral striatum leads to apathy (e.g. Fig 4b, Fig 5b) while the spread of this loss to more dorsal regions of the striatum leads to deficits in inhibiting movements (e.g. Fig 2b, Fig S3). The authors have demonstrated that optogenetic inhibition of D2 neurons in the VLS leads to a reduction in breakpoint in the progressive ratio task. It would be helpful to bolster this claim by optogenetically inhibiting D2 neurons in more dorsal regions to demonstrate (for example) increased premature responding.

We have data demonstrating that the dorsal D2-MSN-optogenetic inhibition resulted in an increase of locomotor activity (please see Figures A and B with legends below), which may account for the idea that the spreading of ablation to more dorsal regions leads to deficits in inhibiting movements. However, we do not want to include these data in revised manuscript because the exploration of the striatal region involving the control of behavioral inhibition is not a major question, and the striatal encoding of such behavior would not be simple. We would like to emphasize again that our major purpose is to demonstrate the role of VLS-D2-MSNs in motivated behaviors.

(A) Illumination of bilateral DS D2-MSNs in D2-ArchT mice. The Black arrow indicates the tip of optic fiber. Scale =1 mm.

(B) The acute optogenetic inhibition of the DS D2-MSNs induced a transient increase of locomotor activity (N = 5 for D2-ArchT and N = 4 for monogenic tetO-ArchT mice, Two-way repeated ANOVA revealed Time \times Group interaction: $F_{9,63}=11.303$, $P<0.001$; One-way repeated ANOVA for bigenic group revealed a main effect of time: $F_{9,36}=19.604$, $P<0.05$; One-way repeated ANOVA for monogenic group found no main effect of Time: $F_{9,27}=2.098$, $P=0.066$; Multiple comparisons with Bonferroni test followed and revealed that photo-inhibition induced a transient increase of locomotor activity: $P<0.05$ when compared to distance in 0-5 min).

Comment #3. *The authors have a bigenic D2-tTA::tetO-ChR2 mouse in their lab. What are the effects of optogenetic activation of D2 neurons in ventrolateral and more dorsal striatum?*

We have data with VLS D2-MSN specific activation by using *Drd2-tTA::tetO-ChR2* mice during the PR task (Please see figures below). Activation shortened the latency to lever press (C, $t_{11} = 2.361$, $P = 0.038$) without altering other parameters (D-G). These results demonstrated the augmented function of VLS D2-MSNs at the initiation of goal-directed behavior.

We would like not to open these data to our revised manuscript because 1) optogenetics-mediated gain-of-function study does not directly supplement the answer to our main question, and 2) we plan to use these data in another paper describing the temporal activities of MSNs during motivated behavior. We hope the reviewer understand our intention.

Comment #4. There are D2-expressing neurons in the cortex, in particular layer 5 neurons in the medial prefrontal cortex. Dopamine in this region is hypothesized to play a role in apathy. It would be straightforward for the authors

to provide an anatomical characterization of DTA mRNA and the loss (or not) of D2 neurons in this region as a time series following DOX off as in Figure 1.

As we replied to the comment #1 from reviewer #3, we summarized the penetrance of DTA induction in D2-DTA mice in **Table S1**. We did not detect *DTA* mRNA in mPFC or IC (**Figure S1F**).

Comment #5. Please plot individual animals (potentially as dots) on all figures with bar graphs (e.g. Figure 2b, 4f, 4g, etc).

We revised the data as reviewer indicated (Figure 2B, 4B-E, 5B-E, 6M-N, and S3A-E).

Reviewer #4

Comment #1. First, the authors propose that ablation of iMSN D2 receptors in the Ventral striatum induces apathy or a state of amotivation. The evidence they sue to support this is that they see a reduction in goal directed behaviour though a decrease in trials and increase in omissions observed 6-7 days after removal of the tet suppression. Similarly the impairment in effort-based instrumental responding (PR) occurred within 3-4 days after removal of tet suppression. At this timepoint they see expression of DTA mRNA, but no alterations in *Drd2* mRNA expression until later (after day 10 of tet off). These timepoints of the behavioral alterations fit within the timeframe of an immunological response (activated microglia), but not necessarily within the timeframe of *Drd2* loss. Therefore, I would interpret their alterations in motivated/goal-directed behaviour would be likely due to an inflammatory response rather than loss of D2 receptors.

As the reviewer #4 pointed out, “the impairment in effort-based instrumental responding (PR) occurred within 3-4 days after removal of tet suppression”. In this time point, we did not observe activated microglia in the striatum (Fig. 1F), suggesting that alterations in motivated behavior would be unlikely due to an inflammatory response.

Around DOX-off days 7, we have to consider the inflammatory response as a confound factor. As we described in the 6th paragraph of Discussion, we evaluated the

effect of inflammatory response on motivated behaviors. To strengthen our evaluation, we added the data showing the alleviation of microglial activation after DOX-restart (**Figure S7**). We would be happy if the reviewer #4 agreed with our thought.

Page 11, line 26 in the 6th paragraph of Discussion:

We also employed DOX-off and restart regimen (Figure 5) in which only a subset of D2-MSNs was ablated and the resultant glial activation was alleviated (**Figure S7**).

Figure S7, related to discussion

Title: Alleviation of microglial activation after DOX-restart.

Legend: *In situ* hybridization for *c-fms* (the marker of microglia) shows resting microglia (DOX-on, left), activated microglia (DOX-off day 7, middle), and resting-like microglia (DOX-off day 7 and restart day 14, right) in the VLS of D2-DTA mice.

Comment #2. *The behavioural data that fits best with the timeframe of loss of D2 receptors is the increased impulsivity and compulsivity on the 3CSRRT (in the supplemental) along with the alterations in locomotor activity. This also supports previous reports of hyperlocomotor activity (Lemos et al., 2016) and increased impulsivity (Dalley et al., 2007) with loss of D2Rs. However, this does not support their hypothesis that decreased D2Rs result in apathy.*

The reviewer #4 might confuse cell dysfunction/cell ablation of D2-MSNs (loss of function of cells) with decreased D2 receptor expression (loss of function of receptors). It is natural that our data with cell dysfunction/ablation study does not fit previous reports with decreased D2 receptor expression.

If not the case, we need to explicitly describe that *Drd2* mRNA disappearance coincided with DTA-mediated cell death. We revised the corresponding sentence as follows.

Page 4, line 21:

In summary, DTA-expressing (*DTA* mRNA-positive) cells were viable for several days, and then cell death occurred (***Drd2* mRNA signal disappearance coincided**).

Comment #3. *If I understand this experiment correctly, to obtain D2-ArchT biogenic mice, they presumably crossed Drd2-tTA mice with TetO-ArchT-EGFP mice (Additional information on this should be in the methods rather than just a reference to the orexin/hypocretin paper in the results).*

As the reviewer expected, we obtained D2-ArchT mice with crossing D2-tTA mice and tetO-ArchT-EGFP mice. To clarify it, we added the term “biogenic” in the method and result sections as follows:

Page 13, line 22 in the method section:

Drd2-tTA::tetO-ArchT-EGFP **biogenic** mice were fed with normal chow (CE-2, CLEA).

Page 8, line 25 in the result section:

We generated **biogenic** animals in which D2-MSNs expressed archaerhodopsin²⁷ (*Drd2-tTA::tetO-ArchT-EGFP*, Figure 6A and **Figure S5B** (Tsunematsu et al., 2013)).

Comment #4. *However, I am unclear how this strategy targets ArchT-EGFP only to the D2R expressed in MSNs and not to all DR2 expressing cells. They indicate that they observed little ArchT-EGFP fluorescence in the VTA dopamine neurons. However, this does not exclude the D2 receptors expressed on glutamatergic inputs or cholinergic inputs to MSNs in the ventral striatum. Presumably inhibition (or light-induced ablation) of these D2-expressing inputs would alter goal-directed behaviour. Can they demonstrate (or further explain) how this targets only MSN D2 receptors?*

Thank you for valuable comments. We provide the characterization of D2-ArchT (**Figure S5**) and the specificity and penetrance of ArchT-EGFP induction (**Table S1**). Please also see our replies on the comment #2 from the reviewer #1 and the comment #1 from the reviewer #3.

Comment #5. Furthermore, could their 3h photostimulation to ablate the D2 expressing cells result in changes in neuroinflammation?

Immediately after opto-ablation of D2-MSNs, massive neuroinflammation occurred but the inflammation alleviated by the behavioral test. Please see our comment #1 and the 6th paragraph of Discussion.

Comment #6. Fig S2c - numbers in the table are way too small to see

We revised the manuscript as the reviewer pointed out (Supplemental Figure S2C).

Comment #7. Data in Fig 3 should be included in the Fig 4, - to save room, move data in 4F,G,H to the supplemental.

We moved data in 4F, G, H to the supplemental (now Figure S4A-C). We would like not to combine Figs 3 and 4 because topics are different.

Comment #8. Line 304 - they are not really looking at reward value, rather reward preference

We rephrased it per the reviewer's comment (page 10, line 7).

Comment #9. Fig S3c - for sensitization they need to test if locomotor activity on day 5 is greater than that on Day in both groups.

We re-analyzed the MAP sensitization data. Two-way repeated ANOVA revealed that there was no Day \times Group interaction ($F_{1,21}=0.190, P=0.677$). We then conducted One-way repeated ANOVA and detected main effects of Drug for control group ($F_1,$

$F_{21}=5.572, P<0.01$) and for DOX off group ($F_{1, 21}=6.976, P<0.01$). The multiple comparisons with Bonferroni method followed and revealed significant increases ($P<0.05$) of locomotor activity between Day 1 and Day 5 for both groups. We described these precise statistical results in our revised supplementary information (**Figure S3F**).

Comment #10. Fig. S4d. They should label the units for the preference score on the y axis.

Done (now Figure S3G).

Reviewers' comments:

Reviewer #1 (Remarks to the Author):

The authors have done a decent job responding the questions and comments I had. The revised manuscript includes more controls and additional data that improves the manuscript and facilitate the interpretation of the results.

My only additional request is that the result section includes a brief statement defining the concept of the ventrolateral striatum to include the lateral part of the nucleus accumbens. Also, a mention that over time the manipulation also causes cell death in the ventral part of the dorsal striatum. All other experiments and answers were already incorporated.

Reviewer #2 (Remarks to the Author):

I thank the authors for their revisions. The new data showing cell death from the optical ablation is convincing.

With respect to my prior points, I'm still stuck on two:

1) The authors responded to my concern that cell death may be an equally viable explanation for their altered ephys responses at day 7 by referencing a schematic (Figure 1F) saying there is no cell death at this point. Can they provide something quantitative on this point? Or write the section in a manner that leaves open this possibility? They conclude that "loss-of-function occurred prior to DTA-mediated cell death," but don't show evidence that there was no cell death at the time of their recordings.

2) I still cannot fully accept the rationale for using the term apathy, and found the authors use of this term at times speculative. For instance, the conclusion sentence of the abstract ends, "thus implicating this circuit in apathy associated with neurodegenerative diseases." No data in their study relates to neurodegenerative diseases, so this type of speculation seems out of place for a conclusion sentence.

In the introduction, the authors cite Levy and Dubois' definition of apathy as a "quantitative reduction of voluntary, goal-directed behaviors", which the authors suggest makes it amenable to study in animals. However, not all reductions in voluntary behavior in humans are caused by, or should be defined as, apathy. Levy and Dubois go on to note that the mechanisms underlying apathy have multiple emotional and psychological sources. I remain unconvinced that mice experience apathy, or that the behavioral tests in this study (which have a long history in the motivation literature) should be described as evidence of apathy.

However, to not get caught in semantics, I would suggest the authors change the term to "apathy-like behavior" and give some description of what exactly they mean by this (reductions in voluntary motor behavior?) if they are set on using this term. Either way, the speculation in the abstract should be kept to a minimum.

Reviewer #3 (Remarks to the Author):

I thank the authors for their response, but note that some concerns were not fully addressed, detailed below:

1) The penetrance and specificity quantification data for the newly generated *Drd2*-tTA mouse are still lacking, and are required to interpret these results. Figure S1C is unclear: is this specificity (% of YFP cells that express *Drd1* or *Drd2*) or penetrance (% of *Drd1* or *Drd2* cells that express YFP)? Both quantities should be depicted. If the presented data is, in fact, specificity, 60% is lower than what is usually accepted for genetic targeting – 90% is a typical minimum percentage. What is the identity of the other 40% of cells, and how do you know they are not responsible for the behavioral effect? The same holds true for Figure S5B.

2) Figure S1F is not sufficiently high resolution to determine whether or not there is DTA mRNA in mPFC or IC. This is a critical question, since the majority of the work in this paper relies on a transgenic approach rather than a spatially restricted viral vector approach. If it is to be believed that the degeneration of VLS D2 neurons leads to apathy, it needs to be shown that D2 neurons elsewhere in the brain are intact. Therefore, it is absolutely essential that 1) the authors provide high-resolution images of mPFC and IC with *Drd2* neurons clearly labeled; 2) the authors show, quantitatively, that there has not been a reduction in this population after DOX-off at several time points up to 20 days; and 3) there is no DTA mRNA in mPFC or IC when assayed at high resolution. Even this is not ideal – D2 neurons elsewhere in the brain could still be mediating the effect (hippocampus, other cortical areas, etc). The cleanest approach would be to use a cre-dependent DTA vector (which exists) in a *Drd2*-cre mouse – this would control for expression everywhere else in the brain. It is difficult to accept a unique role for VLS in apathy without specifically demonstrating that the approach used in this paper does not affect the activity of ALL other *Drd2* cell groups.

3) The authors use a brief 2-second inhibition that starts when the lever is presented, but this inhibition is temporally dissociated from the time during which the behavior is modified. This raises some questions that need to be answered.

a) First, and most important: it is known that periods of optogenetic inhibition can be followed by rebound excitation, and the amount and duration of this rebound excitation may depend on cell type. The authors have shown that a brief 100 ms pulse in a whole cell voltage recording leads to transient inhibition, but they have not demonstrated the effect of this 2-second inhibition on cellular physiology *in vivo*. It is entirely possible that the net effect of this manipulation is enhanced excitability during the bulk of the progressive ratio task, which would muddle the relatively straightforward interpretation presented here.

b) Is there something special about 2 seconds that leads to this behavioral effect? The more obvious experiment would be to turn the laser on at lever presentation and turn it off when the animal has successfully pressed the lever the last time for that trial, or after 5 minutes has passed with no lever press. The results of this kind of experiment would be far easier and more straightforward to interpret, and would be a direct demonstration of the critical role for this cell population in apathy.

Reviewer #4 (Remarks to the Author):

Apathy is lack of interest or lack of an emotional state. In this paper they really did not explore if the mice experienced a lack of emotional state or an indifference to stimuli of positive or negative valence, they only assessed performance on behavioral tasks addressing cognitive responses (attention, impulsive action) and motivation. While lack of self-generated motivational behaviour is a symptom of apathy, they really don't assess whether there is emotional blunting in these animals. On the tasks that they did perform, ie forced swim test and elevated plus-maze, there was not difference in performance between genotypes. The references they offer to support that their definition is 'well accepted' are studies of human Parkinson's patients that also experience poor verbal memory,

emotional blunting, less interest in social activities etc. Furthermore some of their results do not necessarily fit with their definition of apathy (ie increased impulsive action and preservative responding). Therefore, I tend to agree with Reviewer 2 on this point that apathy is a human condition – and if it was possible to model it in mice, they have not done it well here. I am saddened by this current push in today's science to spin results into an anthropomorphic framework for the sake of snappy headlines (ie mice experience loneliness, etc).

With this stated, I think the results of this study are interesting in that they provide a new experimental model testing the contribution of VLS D2-MSNs in behaviour, rigorous, and well controlled and should be published in this journal. However, I really think the 'spin' on apathy is not necessary. I think that it is important that they keep these finding within the context of rodent behaviour. Perhaps in the last paragraph of the discussion they could speculate that these results may fit with certain symptoms of a clinical definition for human apathy, but I really think it is a stretch for the whole paper (introduction and discussion) to be framed around this concept.

My other previous concerns have been addressed.

Below we describe our point-by-point responses to the latest reviewer concerns. The current reviewer comments are shown in italic, and our new responses are shown in blue (our previous responses are shown in regular black text). Revisions in the main text are shown in red.

Reviewer 1

My only additional request is that the result section includes a brief statement defining the concept of the ventrolateral striatum to include the lateral part of the nucleus accumbens.

We added the description that VLS includes the lateral part of the nucleus accumbens.

Page 4, line 35

DTA mRNA expression initiated in the VLS was not limited to the rostral part of the striatum, which included the lateral part of the nucleus accumbens, but spread from the rostral to the caudal part of the striatum

Also, a mention that over time the manipulation also causes cell death in the ventral part of the dorsal striatum

We mentioned above in page 4 line 17 in the previous version:

Drd2 mRNA-negative areas had expanded concentrically by DOX-off day 14 (VLS, ventromedial striatum [VMS], and ventral part of the dorsomedial striatum) (Figures 1C, 1D, and S2) when numerous dead cells were detected.

Reviewer 2

1) The authors responded to my concern that cell death may be an equally viable explanation for their altered ephys responses at day 7 by referencing a schematic (Figure 1F) saying there is no cell death at this point. Can they provide something quantitative on this point? Or write the section in a manner that leaves open this possibility? They conclude that "loss-of-function occurred prior to DTA-mediated cell death," but don't show evidence that there was no cell death at the time of their recordings.

Please see attached figures labeling ssDNA (cell death marker). We did not detect ssDNA-positive cells in the VLS at DOX-off days 7 (n=3 brains).

We just add the phrases in the main text.

Before:

The number of *DTA* mRNA-positive cells was increased at DOX-off day 7, but with no apparent loss of *Drd2* mRNA signal in the corresponding region.

After revision (page 4, line 13):

The number of *DTA* mRNA-positive cells was increased at DOX-off day 7, but with no apparent loss of *Drd2* mRNA signal and with no cell death in the corresponding region.

2) I still cannot fully accept the rationale for using the term *apathy*, and found the authors use of this term at times speculative. For instance, the conclusion sentence of the abstract ends, "thus implicating this circuit in *apathy* associated with neurodegenerative diseases." No data in their study relates to neurodegenerative diseases, so this type of speculation seems out of place for a conclusion sentence.

In the introduction, the authors cite Levy and Dubois' definition of *apathy* as a "quantitative reduction of voluntary, goal-directed behaviors", which the authors suggest makes it amenable to study in animals. However, not all reductions in voluntary behavior in humans are caused by, or should be defined as, *apathy*. Levy and Dubois go on to note that the mechanisms underlying *apathy* have multiple emotional and psychological sources. I remain unconvinced that mice experience *apathy*, or that the behavioral tests in this study (which have a long history in the motivation literature) should be described as evidence of *apathy*.

However, to not get caught in semantics, I would suggest the authors change the term to "*apathy-like behavior*" and give some description of what exactly they mean by this (reductions in voluntary motor behavior?) if they are set on using this term. Either way, the speculation in the abstract should be kept to a minimum.

We agree with your concern. We remove the term *apathy* from the title. We carefully edit the abstract, the introduction, and the last paragraph of the discussion.

Reviewer 3

1) *The penetrance and specificity quantification data for the newly generated Drd2-tTA mouse are still lacking, and are required to interpret these results. Figure S1C is unclear: is this specificity (% of YFP cells that express Drd1 or Drd2) or penetrance (% of Drd1 or Drd2 cells that express YFP)? Both quantities should be depicted.*

We will add the information in the Figure S1 and S5 legends and revise y-axis label (Figures S1F and S5B).

If the presented data is, in fact, specificity, 60% is lower than what is usually accepted for genetic targeting – 90% is a typical minimum percentage.

In case of tetracycline-controllable gene induction system, 60% is high. The penetrance of tTA-mediated gene induction is dependent of the nature of tetO-line. For example, most famous line CaMKII-tTA (made by Mark Mayford and Eric Kandel) shows 0 % penetrance of tTA-mediated gene induction in cortical neurons in some plasmid tetO lines (Kellendonk et al, Neuron 2006; Tanaka et al., Cell Rep 2012). On the other hand, the penetrance of tTA expression by D2 promoter is over 90%. More importantly, specificity to D2-MSNs is over 90%.

What is the identity of the other 40% of cells, and how do you know they are not responsible for the behavioral effect? The same holds true for Figure S5B.

The supplemental figure shows penetrance, not specificity (as this comment seems to assume). The other 40% of cells are GFP-negative/D2-positive cells. GFP-positive/D1-positive cell are rare (less than 3%, Figure S1C and Figure S5B), which is unlikely to affect the main effect.

2) *Figure S1F is not sufficiently high resolution to determine whether or not there is DTA mRNA in mPFC or IC. This is a critical question, since the majority of the work in this paper relies on a transgenic approach rather than a spatially restricted viral vector approach. If it is to be believed that the degeneration of VLS D2 neurons leads to apathy, it needs to be shown that D2 neurons elsewhere in the brain are intact. Therefore, it is absolutely essential that:*

(1) the authors provide high-resolution images of mPFC and IC with Drd2 neurons clearly labeled;

We provided high-resolution images of mPFC and IC with *Drd2* neurons in Figure S3B.

B

(2) the authors show, quantitatively, that there has not been a reduction in this population after DOX-off at several time points up to 20 days

We provided the cell number for both cortices below.

Relative number of Drd2 neurons	DOX on	DOX-off day 10	DOX-off day 20
in mPFC (%)	100	108	100
in IC (%)	100	109	105

(n=3)

This seems to be meaningless because we never find DTA mRNA in mPFC or IC. We think that it is not necessary to provide this information even in the supplementary figure.

(3) *there is no DTA mRNA in mPFC or IC when assayed at high resolution.*

We provided high-resolution images of DTA mRNA ISH in Figure S3A.

Even this is not ideal – D2 neurons elsewhere in the brain could still be mediating the effect (hippocampus, other cortical areas, etc). The cleanest approach would be to use a cre-dependent DTA vector (which exists) in a Drd2-cre mouse – this would control for expression everywhere else in the brain. It is difficult to accept a unique role for VLS in apathy without specifically demonstrating that the approach used in this paper does not affect the activity of ALL other Drd2 cell groups.

One of our key points is that the optogenetic ablation and silencing experiments both target only VLS, and these studies confirm a unique role for VLS (Fig 6). The viral

approach suggested by the reviewer could have been used to accomplish the same objective. However, we believe the optogenetic approach is preferable because it demonstrates that the same phenotype can be produced via both silencing and ablation. The virus experiment would not provide any additional information beyond what was provided by the optogenetic experiments.

3) *The authors use a brief 2-second inhibition that starts when the lever is presented, but this inhibition is temporally dissociated from the time during which the behavior is modified. This raises some questions that need to be answered.*

a) *First, and most important: it is known that periods of optogenetic inhibition can be followed by rebound excitation, and the amount and duration of this rebound excitation may depend on cell type. The authors have shown that a brief 100 ms pulse in a whole cell voltage recording leads to transient inhibition, but they have not demonstrated the effect of this 2-second inhibition on cellular physiology in vivo. It is entirely possible that the net effect of this manipulation is enhanced excitability during the bulk of the progressive ratio task, which would muddle the relatively straightforward interpretation presented here.*

We believe our comment #3 from the previous response should have addressed this concern. It is possible that 2-second ArchT opening caused enhanced excitability. However, as we showed in the previous response, ChR2-mediated activation of VLS D2-MSNs produces a behavioral effect that is totally different from the effect of ArchT-mediated silencing. This means that the net effect of ArchT is unlikely to be excitation. Furthermore, the effects of ArchT inhibition are identical to the effects of D2-MSN ablation (Figure 6), also suggesting that the predominant effect of ArchT is neural silencing.

b) *Is there something special about 2 seconds that leads to this behavioral effect? The more obvious experiment would be to turn the laser on at lever presentation and turn it off when the animal has successfully pressed the lever the last time for that trial, or after 5 minutes has passed with no lever press. The results of this kind of experiment would be far easier and more straightforward to interpret, and would be a direct demonstration of the critical role for this cell population in apathy.*

We measured the VLS D2-MSNs population activity by using fiber photometry, and identified that VLS D2-MSN activity showed the Ca^{2+} surge for about 2 seconds immediately after the trial start cue. We will show these data below, but we prefer not to include these data in the manuscript as they are part of another paper.

Fig: Temporal changes of Ca^{2+} signals in VLS D2-MSNs aligned to the timing of trial start during operant behaviors. Red bar indicates the period (about 2 seconds) with statistical difference ($p < 0.01$)

Reviewer 4

Apathy is lack of interest or lack of an emotional state. In this paper they really did not explore if the mice experienced a lack of emotional state or an indifference to stimuli of positive or negative valence, they only assessed performance on behavioral tasks addressing cognitive responses (attention, impulsive action) and motivation. While lack of self-generated motivational behaviour is a symptom of apathy, they really don't assess whether there is emotional blunting in these animals. On the tasks that they did perform, ie forced swim test and elevated plus-maze, there was not difference in performance between genotypes. The references they offer to support that their definition is 'well accepted' are studies of human Parkinson's patients that also experience poor verbal memory, emotional blunting, less interest in social activities etc. Furthermore some of their results do not necessarily fit with their definition of apathy (ie increased impulsive action and preservative responding). Therefore, I tend to agree with Reviewer 2 on this point that apathy is a human condition – and if it was possible to model it in mice, they have not done it well here. I am saddened by this current push in today's science to spin results into an anthropomorphic framework for the sake of snappy headlines (ie mice experience loneliness, etc).

With this stated, I think the results of this study are interesting in that they provide a new experimental model testing the contribution of VLS D2-MSNs in behaviour, rigorous, and well controlled and should be published in this journal. However, I really think the 'spin' on apathy is not necessary. I think that it is important that they keep these finding within the context of rodent behaviour. Perhaps in the last paragraph of the discussion they could speculate that these results may fit with certain symptoms of a clinical definition for human apathy, but I really think it is a stretch for the whole paper (introduction and discussion) to be framed around this concept.

We agree with this concern. We keep our finding within the context of rodent behavior and carefully edit the abstract, the introduction, and the last paragraph of the discussion.

REVIEWERS' COMMENTS:

Reviewer #2 (Remarks to the Author):

The authors have addressed all of my concerns, thank you.

Reviewer #3 (Remarks to the Author):

I thank the authors for their efforts to address my concerns. I am satisfied. Some points for consideration below:

1) Thank you for providing the specificity data for the newly generated Drd2-tTA mice. I am still somewhat concerned about specificity, particularly since this is a new mouse line that will be available to the community. Two reasons for this concern: 1) Drd1 and Drd2 antibodies are notoriously unreliable, and 2) ChR2/ArchT-EYFP is a membrane-bound fluorophore, which has a lower accuracy for specificity determination than a cell filling construct like EYFP alone (it can be hard to differentiate soma membrane from processes). However, the authors have met the current standard for the field so this will not preclude publication.

2) Thank you for providing the high resolution images of DTA mRNA. I am convinced there is no expression in cortex. And you are right that the ArchT optogenetic experiments resolve this concern. Keep in mind for future experiments that the bigenic approach (Drd2-tTA::tetO-ChR2-EGFP) should NOT be used for ChR2 experiments, since stimulation in the VLS will lead to antidromic activation of other ChR2-expressing D2R cells that send axons to the VLS. A viral vector injection in VLS would be a better approach as it removes this concern.

3) Thank you for the demonstration that ArchT-expressing cells do not show rebound excitation, and for the detailed explanation. I am convinced. I like the fiber photometry data and agree that it can be held for subsequent paper. If the authors are so inclined, for this future paper it would be interesting to record fiber photometry data while optogenetically inhibiting the Drd2 population – a prediction would be that these two seconds of inhibition would be enough to prevent any rise of activity at all, which would be interesting because it would demonstrate that precisely timed inhibition can have long lasting network effects.

Below we describe our point-by-point responses (**Gothic**) to Reviewer 3 comments (*italic*)

1) Thank you for providing the specificity data for the newly generated Drd2-tTA mice. I am still somewhat concerned about specificity, particularly since this is a new mouse line that will be available to the community. Two reasons for this concern: 1) Drd1 and Drd2 antibodies are notoriously unreliable, and 2) ChR2/ArchT-EYFP is a membrane-bound fluorophore, which has a lower accuracy for specificity determination than a cell filling construct like EYFP alone (it can be hard to differentiate soma membrane from processes). However, the authors have met the current standard for the field so this will not preclude publication.

We agree that almost all commercially available D1 and D2 antibodies are not good. However, antibodies generated by Dr. Watanabe (he is a co-author in this article) are very good. He validated the specificity of Drd1 and Drd2 antibodies by using knockout mouse brains. These antibodies can be obtained through Frontier Institute at Japan.

As you know, Drd1, Drd2, and opsin-EYFP also localize in the endoplasmic reticulum in addition to the plasma membrane. Therefore, we are able to determine the cell identity by their intracellular (in ER) fluorophore locations. Indeed, Dr. Watanabe's group has successfully determined the specificity of cell type using these antibodies and counting method (Narushima et al., 2006; Uchigashima et al., 2007; Uchigashima et al., 2016). We believe that our method for cell type determination can be the best way so far.

Narushima, M., Uchigashima, M., Hashimoto, K., Watanabe, M., Kano, M. (2006) Depolarization-induced suppression of inhibition mediated by endocannabinoid at synapses from fast-spiking interneurons to medium spiny neurons in the striatum. *Eur. J. Neurosci.* 24:2246-2252.

Uchigashima M, Narushima M, Fukaya M, Katona I, Kano M, Watanabe M: Subcellular arrangement of molecules for 2-arachidonoyl-glycerol-mediated retrograde signaling and its physiological contribution to synaptic modulation in the striatum. *J. Neurosci.*, 27:3663-3676, 2007.

Uchigashima M, Ohtsuka T, Kobayashi K, Watanabe M. Striatal dopamine synapses are neuroligin-2-mediated contact between dopaminergic presynaptic and GABAergic postsynaptic structures. *Proc Natl Acad Sci USA*, 113:4206-4211, 2016.

2) Thank you for providing the high resolution images of DTA mRNA. I am convinced there is no expression in cortex. And you are right that the ArchT optogenetic experiments resolve this concern. Keep in mind for future experiments that the bigenic approach (Drd2-tTA::tetO-ChR2-EGFP) should NOT be used for ChR2 experiments, since stimulation in the VLS will lead to antidromic activation of other ChR2-expressing D2R cells that send axons to the VLS. A viral vector injection in VLS would be a better approach as it removes this concern.

We do not include behavioral data obtained with *Drd2-tTA::tetO-ChR2-EYFP*

mice in this study; therefore, we do not show the detail of ChR2-EYFP expression profile. Please understand that ChR2-EYFP induction patterns are identical to that of ArchT-EGFP because both tetO-cassettes were inserted into the same beta-actin locus (Tanaka et al. 2012; Tsunematsu et al., 2013), which indicates the same tTA-mediated gene induction patterns (no cortical ChR2 expression). Therefore, we fortunately can use *Drd2*-tTA::tetO-ChR2-EYFP mice for striatal photoactivation experiments.

3) Thank you for the demonstration that ArchT-expressing cells do not show rebound excitation, and for the detailed explanation. I am convinced. I like the fiber photometry data and agree that it can be held for subsequent paper. If the authors are so inclined, for this future paper it would be interesting to record fiber photometry data while optogenetically inhibiting the Drd2 population – a prediction would be that these two seconds of inhibition would be enough to prevent any rise of activity at all, which would be interesting because it would demonstrate that precisely timed inhibition can have long lasting network effects.

Thank you for your comment. We would like to include your comment into our preparing paper using fiber photometry.